# Distinct responses to reduplicated chromosomes require distinct Mad2 responses

**Benjamin M Stormo[1], Donald T Fox[1,2]\***

[1]Department of Cell Biology, Duke University Medical Center, Durham, United States; [2]Department of Pharamacology and Cancer biology, Duke University Medical Center, Durham, United States

**Abstract** Duplicating chromosomes once each cell cycle produces sister chromatid pairs, which separate accurately at anaphase. In contrast, reduplicating chromosomes without separation frequently produces polytene chromosomes, a barrier to accurate mitosis. Chromosome reduplication occurs in many contexts, including: polytene tissue development, polytene tumors, and following treatment with mitosis-blocking chemotherapeutics. However, mechanisms responding to or resolving polyteny during mitosis are poorly understood. Here, using *Drosophila*, we uncover two distinct reduplicated chromosome responses. First, when reduplicated polytene chromosomes persist into metaphase, an anaphase delay prevents tissue malformation and apoptosis. Second, reduplicated polytene chromosomes can also separate prior to metaphase through a spindle-independent mechanism termed Separation-Into-Recent-Sisters (SIRS). Both reduplication responses require the spindle assembly checkpoint protein Mad2. While Mad2 delays anaphase separation of metaphase polytene chromosomes, Mad2's control of overall mitotic timing ensures efficient SIRS. Our results pinpoint mechanisms enabling continued proliferation after genome reduplication, a finding with implications for cancer progression and prevention.

**\*For correspondence:** don.fox@duke.edu

## Introduction

Regulating mitotic chromosome structure is critical to preventing genomic instability (*Gordon et al., 2012*; *Pfau and Amon, 2012*). During mitosis, chromatids associate in sister pairs, which facilitates their bi-orientation and subsequent segregation to opposite spindle poles. A frequently occurring and long-recognized departure from this paired chromosome structure occurs when the genome reduplicates without chromatid separation (hereafter: genome reduplication). Following a single extra S-phase, cells frequently form diplochromosomes: four sister chromatids conjoined at centromeres (*White, 1935*). A more general term for chromosomes formed by any degree of genome reduplication without chromatid separation is 'polytene' (*Painter, 1934*; *Zhimulev et al., 2004*).

While incompletely understood, it is appreciated that multiple layers of physical connections tightly intertwine the multiple sister chromatids of polytene chromosomes. These connections likely include cohesins (*Cunningham et al., 2012*; *Pauli et al., 2010*) as well as topological entanglements that can be removed by Condensin II activity (*Bauer et al., 2012*; *Smith et al., 2013*; *Wallace et al., 2015*). Additionally, recurring regions of DNA under-replication occur between chromatids in some polytene cells (*Beliaeva et al., 1998*; *Gall et al., 1971*; *Hannibal et al., 2014*; *Nordman et al., 2011*; *Yarosh and Spradling, 2014*) whereas DNA replication is more complete in others (*Dej and Spradling, 1999*; *Fox et al., 2010*). In addition to connections between sister chromatids, another layer of chromosome association - pairing between homologs - also occurs in some polytene cells. This pairing results in polyploid/polytene cells that exhibit only the haploid number of distinct

**eLife digest** Before a cell divides, it duplicates all its genetic information, which is stored on chromosomes. Then, each chromosome evenly divides into two new cells so that each cell ends up with identical copies of the genetic information. Because the cellular machinery that evenly divides chromosomes is built to recognize chromosomes that were duplicated exactly once, it is important to maintain this pattern of alternating one round of duplication with one round of division. Cells that instead duplicate their chromosomes more than once can make mistakes during division that are associated with diseases such as cancer.

Chromosomes with extra duplications are present in normal tissues such as the placenta of mammals. They can also occur in human diseases and may even result from chemotherapy treatment. However, we know almost nothing about how cells respond to these problematic chromosomes when dividing.

By studying cells from the *Drosophila melanogaster* species of fruit fly, Stormo and Fox discovered two distinct ways in which cells respond to extra chromosome duplications. One response occurs in cells that were experimentally engineered to undergo an extra chromosome duplication. These cells delay division so that the chromosome separation machinery can somehow adapt to the challenge of separating more than two chromosome copies at once. The second response occurs in cells that naturally undergo extra chromosome duplications before division. In these cells, Stormo and Fox discovered a new type of chromosome separation, whereby the extra chromosome copies move apart from each other before cell division. In doing so the chromosomes can better interact with the chromosome separation machinery during division.

Stormo and Fox also found that a protein named Mad2 is important in both responses, and gives the cell enough time to respond to extra chromosome copies. Without Mad2, the separation of chromosomes with extra duplications is too hasty, and can lead to severe cell division errors and cause organs to form incorrectly.

Having uncovered two new responses that cells use to adapt to extra chromosomes, it will now be important to find other proteins like Mad2 that are important in these events. Understanding these processes and the proteins involved in more detail could help to prevent diseases that are associated with extra chromosomes.

chromosomes (*Metz, 1916*; *White, 1954*). Given these multiple physical connections between polytene chromatids, mitosis in polytene cells is considered 'ill-advised for mechanical reasons' (*Edgar and Orr-Weaver, 2001*). Indeed, separation of polytene diplochromosomes at anaphase causes chromosome mis-segregation (*Vidwans et al., 2002*).

Given the association of polytene chromosomes with mitotic errors, it is not surprising that these structures are often associated with aberrant development and disease. Polytene chromosomes have been observed in cells from spontaneous human abortions (*Therman et al., 1978*), in muscular dystrophy patients (*Schmidt et al., 2011*), in a variety of tumors (*Biesele and Poyner, 1943*; *Erenpreisa et al., 2009*; *Therman et al., 1983*) and can also precede tumor formation in mice (*Davoli and de Lange, 2012*). Polytene chromosomes also occur after treatment with currently used anti-mitotic chemotherapeutics such as those that inhibit Topoisomerase II (*Cantero et al., 2006*; *Sumner, 1998*). Disruption of numerous other processes crucial for mitosis, including spindle formation (*Goyanes and Schvartzman, 1981*; *Takanari et al., 1985*) sister chromatid cohesion (*Wirth et al., 2006*) or genome integrity control (*Davoli et al., 2010*) also cause genome reduplication and polyteny. Thus, polytene chromosomes, a source of mitotic instability, are a conserved and common outcome of ectopic genome reduplication.

To understand how cells adapt the cell cycle machinery to the challenge of segregating the intertwined polytene chromatids found in genome-reduplicated cells, naturally occurring models of this problem can prove useful. Programmed genome reduplication cycles of successive S-phase without M-phase (endocycles, *Edgar et al., 2014*; *Fox and Duronio, 2013* see nomenclature) produce polytene chromosomes in many plant, insect, and mammalian species, including humans (*Zhimulev et al., 2004*; *Zybina et al., 1996*). However, many cells with programmed genome

reduplication do not subsequently divide, preventing study of how nature has circumvented the issue of segregating polytene chromosomes. In contrast, we previously demonstrated that rectal papilla (hereafter: papillar cells), ion-absorbing structures in the *Drosophila* hindgut, are built entirely by mitosis of endocycled cells (*Fox et al., 2010*; *Schoenfelder et al., 2014*). Surprisingly, we never observed polytene chromosomes in hundreds of papillar metaphases (*Fox et al., 2010*; *Schoenfelder et al., 2014*), suggesting papillar cells are programmed to either avoid or eliminate polyteny and its associated mitotic defects. Interestingly, previous studies suggest that polyteny can be at least partially undone without anaphase in both normal and tumorous tissue (*Dej and Spradling, 1999*; *Grell, 1946*; *Levan and Hauschka, 1953*). Thus, in some cases, polyteny may be actively regulated or eliminated.

Taken together, the potential negative impact of genome reduplication on mitotic chromosome structure is clear. However, the responses that enable either developing or tumorous cells to continue dividing after reduplication, despite profound chromosome structure changes, remain unclear. Here, using *Drosophila* tissue models of both ectopic and naturally occurring genome reduplication, we uncover two distinct cellular responses to reduplicated chromosomes. Both reduplication responses require the conserved spindle assembly checkpoint (SAC) protein Mad2, which inhibits the Anaphase Promoting Complex to both delay anaphase in response to unattached or tensionless kinetochores and to also regulate overall mitotic timing from nuclear envelope breakdown (NEBD) to anaphase onset (*London and Biggins, 2014*; *Musacchio, 2015*). In reduplicated cells that retain polytene chromosomes at metaphase, we show Mad2 is involved in a SAC wait-anaphase response. This anaphase delay does not fully prevent the mitotic errors and the resulting aneuploidy associated with mitosis of polytene chromosomes, but it substantially reduces apoptosis, tissue malformation, and organismal death. In contrast to this wait-anaphase response, we also define a second response in reduplicated cells that actively eliminates polyteny before anaphase. In this response, polytene chromosomes undergo a dynamic, spindle-independent process we term Separation Into Recent Sister chromatid pairs (SIRS), which eliminates any trace of polyteny before anaphase. Unlike mitosis with metaphase polytene chromosomes, mitosis with SIRS does not trigger a Mad2-dependent anaphase delay. Yet, we find Mad2 promotes efficient SIRS by allowing sufficient time between nuclear envelope breakdown and anaphase, which allows polytene chromosomes to separate into conventional mitotic sister chromatid pairs. Our results therefore define two distinct responses to reduplicated chromosomes, each of which depends on a distinct Mad2 response.

## Results

To understand the mechanisms employed by cells with reduplicated chromosomes, we took advantage of accessible developmental models and in vivo genetic tools in *Drosophila*. While ectopic genome reduplication was previously established to generate polytene diplochromosomes and subsequent mitotic errors in *Drosophila*, the effects were examined within the time-frame of the terminal embryonic mitotic cell cycle (*Vidwans et al., 2002*). Thus, the long-term effects of ectopic genome reduplication on cell viability and tissue development, and key molecular regulation of reduplicated chromosomes has remained unexplored. In parallel, our development of rectal papillae as a non-ectopic, naturally occurring model of mitosis after genome reduplication enabled us to also study how cells programmed to undergo genome reduplication can regulate polytene chromosome structure during mitosis.

### Ectopic genome reduplication yields polyteny and continued aneuploid cell division

We first ectopically induced genome reduplication in proliferating tissues of developing larvae by transiently re-programming mitotic cycles to endocycles. *fizzy-related (fzr, mammalian Cdh1)* plays a conserved role in endocycles by targeting the anaphase promoting complex to destroy the mitotic Cyclins A, B, and B3 (*Larson-Rabin et al., 2009*; *Sigrist and Lehner, 1997*). *fzr* overexpression was previously shown to transform mitotic cycles into endocycles (*Sigrist and Lehner, 1997*). To transiently induce endocycles, we used a brief heat shock (HS) pulse to express ectopic *fzr (HS>fzr,* *Figure 1A*). Using the cell cycle marker system Fly-FUCCI (*Zielke et al., 2014*; *Figure 1B*) we find that pulsed *fzr* overexpression temporarily eliminates expression of the S/G2/M mRFP-CyclinB reporter in wing imaginal disc cells (*Figure 1C vs. C', D*). This same population of mRFP-CyclinB-

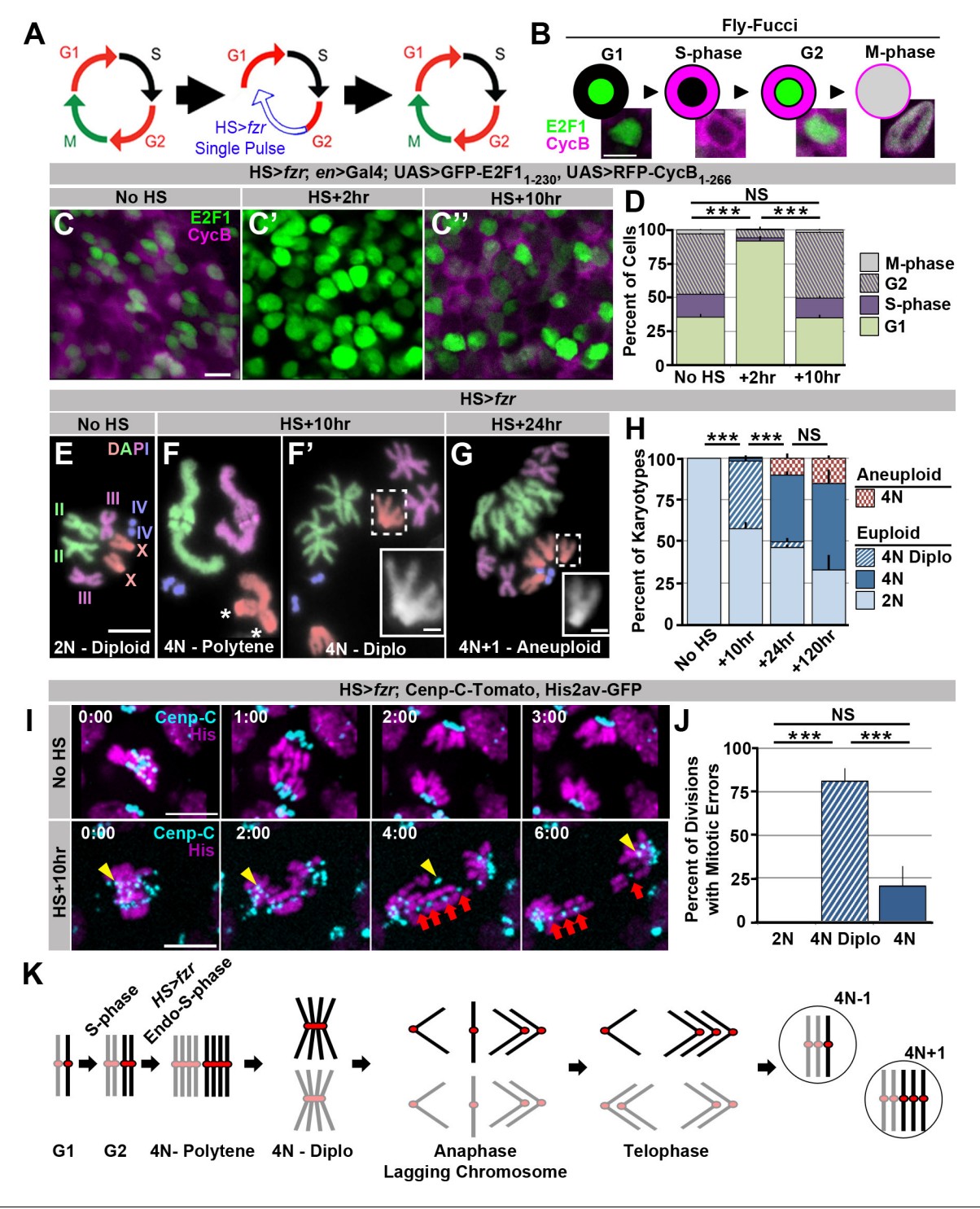

**Figure 1.** Induced genome reduplication in wing progenitors leads to polytene diplochromosomes and aneuploidy. (A) A model for the cell cycle progression following *fizzy-related (HS>fzr)* overexpression in a mitotically cycling tissue. Cells progress directly from G2 into G1 without an intervening mitosis, resulting in an additional S-phase. (B) A diagram depicting the Fly-FUCCI system in each stage of the cell cycle, and representative images of wing imaginal disc cells in each cell cycle state. GFP-E2F1$_{1-230}$ (green) is nuclear during G1 and G2 and fills the cell during mitosis. RFP-CycB$_{1-266}$ (magenta) is cytoplasmic during S-phase and G2 and fills the cell during mitosis. (C) Representative micrographs of the wing imaginal disc pouch expressing UAS Fly-FUCCI under the control of *engrailed-Gal4* in the absence of *HS>fzr* expression (No HS, C) as well as +2 hr (C') and +10 hr (C'') after a 60-min heat shock to induce *HS>fzr* expression. GFP-E2F1$_{1-230}$ is in green, RFP-CycB$_{1-266}$ is in magenta. (D) The percentage of cells in G1, S, G2, and M based on Fly-FUCCI expression prior to *HS>fzr* expression (No HS), +2 hr and +10 hr after a 60-min heat shock to induce *fzr* expression. Stacked
*Figure 1 continued on next page*

*Figure 1 continued*

bars represent mean plus standard error of the mean (+S.E.M.), ***p<0.001, NS = p>0.05, t-test. Data are an average of three replicates with at least 5 animals per replicate and at least 50 cells counted per animal. (E) Representative karyotypes from a mitotic *HS>fzr* wing imaginal disc cells without heat shock. Chromosomes are pseudocolored according to each chromosome type and numbered. Prior to *HS>fzr* expression cells display a normal diploid karyotype. Tissue was incubated in colcemid for 30 min to enrich for mitotic cells. (F) Representative karyotypes from mitotic *HS>fzr* wing imaginal cells 10 hr after a 60-min heat shock. Chromosomes are pseudocolored according to the type as in *Figure 1E*. Transiently, closely aligned polytene chromosomes are seen when chromosomes first condense after genome reduplication (F). Asterisks indicate the 2 groups of homologous centromeres of the X-chromosome. Diplochromosomes, characterized by the attachment of four centromeres of each sister chromatid (F′ see inset), are seen at the first metaphase after genome reduplication. Tissue was incubated in colcemid for 30 min to enrich for mitotic cells. (G) Representative karyotype of a mitotic *HS>fzr* cell 24 hr after a 60 minheat shock, colored according to type as in *Figure 1E*. Aneuploid cells are observed at 24 hr after heat shock, during the second metaphase after genome reduplication, which follows the division of diplochromosomes. Tissue was incubated in colcemid for 30min to enrich for mitotic cells. (H) The percentage of wing imaginal disc karyotypes classified as euploid/diploid, euploid/tetraploid, euploid/diplo-tetraploid, or aneuploid/tetraploid prior to heat shock (No HS), or +10 hr, +24 hr, or +120 hr after a 60-min heat shock. Stacked bars represent Mean (+S.E.M.), ***=p<0.001, NS = p>0.05, t-test. Data are an average of 3 replicates with at least 50 karyotypes per replicate. (I) Representative time-lapse of a diploid wing imaginal disc cell dividing prior to *HS>fzr* expression (No HS) and a tetraploid cell with diplochromosomes dividing 10 hr after a 60-min heat shock to induce *HS>fzr* expression (HS +10 hr). Yellow arrowhead shows a single lagging kinetochore. Red arrows highlight a single diplochromosome that segregates its chromatids in a 3:1 fashion. Cenp-C-Tomato showing kinetochores in cyan, His2av-GFP showing DNA in magenta. Time represents min from the last frame prior to anaphase. (J) The percentage of lagging chromosomes in diploid cells, in tetraploid cells with diplochromosomes (4N Diplo), and in tetraploid cells without diplochromosomes (4N) after *HS>fzr* expression. Bars represent averages (+S.E.M.) between animals with at least five animals per condition. ***p<0.001, NS = p>0.05, t-test. (K) A model for a cell cycle that results in aneuploid daughter cells showing only the two homologs of a single chromosome for simplicity. The two homologs are shown in black and gray with a red centromere. Chromatids are replicated in S-phase and then reduplicated following a heat shocked induced endocycle. This results in polytene chromosomes. Diplochromosomes are seen as the genome-reduplicated cells progress into metaphase. At anaphase, diplochromosome segregation frequently produces lagging chromatids, which can segregate erroneously resulting in aneuploidy. Scale bars represent 5 µm, except in insets in F′ and G where it represents 1 µm.

The following figure supplement is available for figure 1:

**Figure supplement 1.** Supporting data regarding the effect of *HS>fzr* on imaginal discs and brains.

negative cells continues to express the G2/M/G1 GFP-E2F1 reporter, but in greater proportion (*Figure 1C vs. C′, D*). Together, these data suggest *HS>fzr* promotes G1 accumulation (91% of cells compared to 36% in controls, *Figure 1B–D*). To test whether this G1 accumulation is due to direct conversion of G2 cells to G1, as opposed to an acceleration of the cell cycle through G2/M, we stained for the mitotic marker Phospho-Histone H3 at several time points after pulsed *fzr* expression. For up to 7 hr after *fzr* overexpression, there is essentially no mitosis in the wing imaginal disc, whereas wing cells in heat shocked wild type flies continue to divide after heat shock (*Figure 1—figure supplement 1A,B*). Based on previous studies of *fzr* function and our FUCCI and Phospho-Histone H3 data, we conclude that pulsed *fzr* expression converts G2 cells to a G1 state by eliminating mitotic cyclins (*Figure 1A*).

To determine if the G2 cells re-programmed to G1 proceed through a second genome duplication, we examined mitotic chromosome number when mitosis of *HS>fzr* tissue first resumes (ten hours after heat shock). At this time-point, we observe frequent tetraploidy (41% of mitotic cells, equivalent to 93% of all G2 cells prior to heat shock, *Figure 1E v. F,F′,H*). We obtained similar results when examining the results of *fzr* overexpression in diploid brain progenitors (*Figure 1—figure supplement 1C*). These results confirm our ability to induce genome reduplication in normally diploid tissues. We also examined chromosome structure in our induced tetraploid cells. When *HS>fzr*-induced tetraploid DNA first condenses post-heat shock, all chromatids of each chromosome type are closely aligned in a polytene configuration, as evidenced by having the haploid number of distinguishable chromosomes (four in females, *Figure 1F*). Frequently, we observe un-pairing of the homologous groups of centromeres within each polytene chromosome (*Figure 1F*, asterisk, also see discussion). Later, at the first metaphase, homologous chromosomes of each polytene are now completely separated, but the four centromeres of each group of sister chromatids remain conjoined within diplochromosomes (observed for 96% of tetraploid cells, *Figure 1F′* inset, H). Thus, *HS>fzr* induces ectopic genome reduplication, resulting in tetraploid cells with metaphase polytene diplochromosomes.

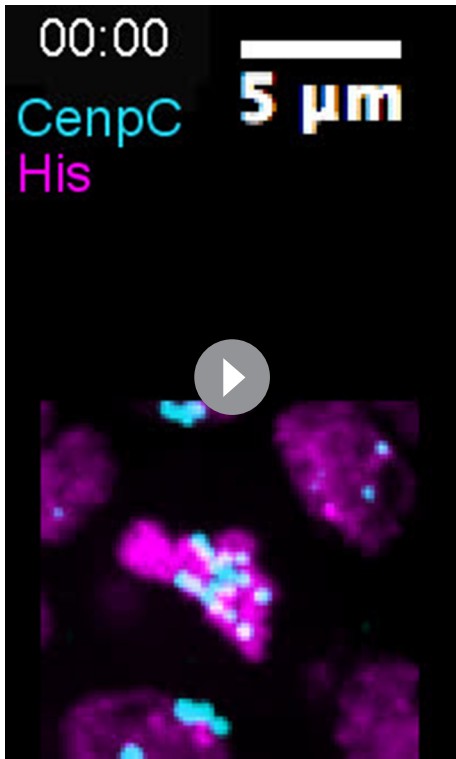

**Video 1.** This video accompanies *Figure 1I*. Live imaging of a diploid HS>*fzr* wing imaginal disc cell dividing prior to *HS>fzr* expression showing His2av-GFP in magenta to label DNA, and Cenp-C-Tomato in cyan to label kinetochores. No mitotic errors are detected. Time Indicates minutes to the last frame of metaphase, scale bars represent 5 μm.

**Video 2.** This video accompanies *Figure 1I*. Live imaging of a tetraploid *HS>fzr* wing imaginal disc cell with diplochromosomes 10 hr after a 60-min heat shock to induce *HS>fzr* expression with His2av-GFP in magenta to label DNA, and Cenp-C-Tomato in cyan to label kinetochores. Lagging chromosomes are evident. Time Indicates minutes to the last frame of metaphase, scale bars represent 5 μm.

We next examined the mitotic fidelity of cells with diplochromosomes by two independent means: chromosome karyotype analysis and live imaging. By examining the metaphase chromosomes of the division immediately following diplochromosome division, we could detect whether aneuploidy results from diplochromosome segregation. Following the division of cells with diplochromosomes, we observe tetraploid-aneuploid cells with one or two extra or missing chromosomes (8.6% of mitotic cells). In these cells, diplochromosomes are no longer present and instead chromatids are found in distinct sister pairs (*Figure 1G*). This suggests that during or after anaphase of the first post-reduplication division, diplochromosomes can separate into individual chromatids. Further, these diplochromosome divisions can produce aneuploid daughter cells, many of which continue to divide (*Figure 1G,H,K*, *Figure 1—figure supplement 1C*).

We also live imaged mitosis of wing imaginal disc and brain progenitor (neuroblasts and ganglion mother) cells, both with and without ectopic genome reduplication. In addition to using a histone marker to observe chromosomes, we used the Cenp-C-Tomato marker to observe kinetochores. Control diploid cells divide without errors (*Figure 1I* No HS, J, *Video 1*). In contrast, most (80%) tetraploid divisions with diplochromosomes exhibit lagging chromosomes, DNA bridges, or both (*Figure 1I* HS+10 hr, J, *Video 2*). In our live imaging, diplochromosomes were identifiable as quartets of centromeres and their associated chromosome arms in very close proximity. In some of these divisions we clearly observe four chromatids of a diplochromosome quartet segregating 3:1 (in agreement with prior work in embryos by *Vidwans et al., 2002*, suggesting incomplete or imprecise sister chromatid disjunction is the cause of chromosome gains and losses (*Figure 1I*, *Figure 1—figure supplement 1D*, *Video 2*). Mitotic errors in the first division of *HS>fzr* cells appear to result primarily from diplochromosomes and not tetraploidy itself, as tetraploid cells in the subsequent

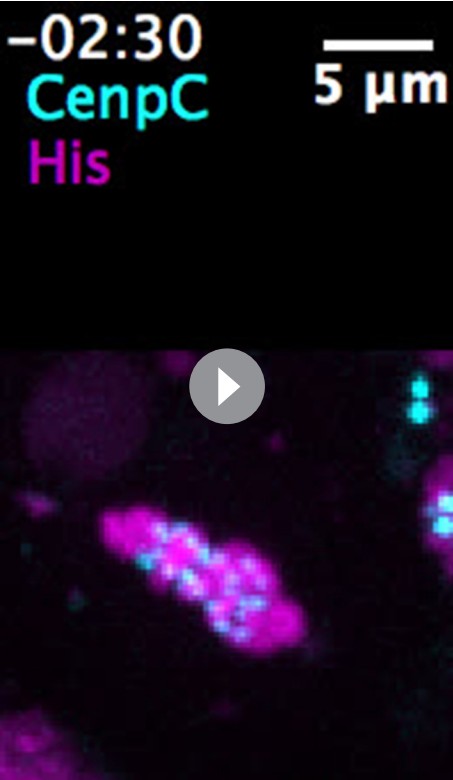

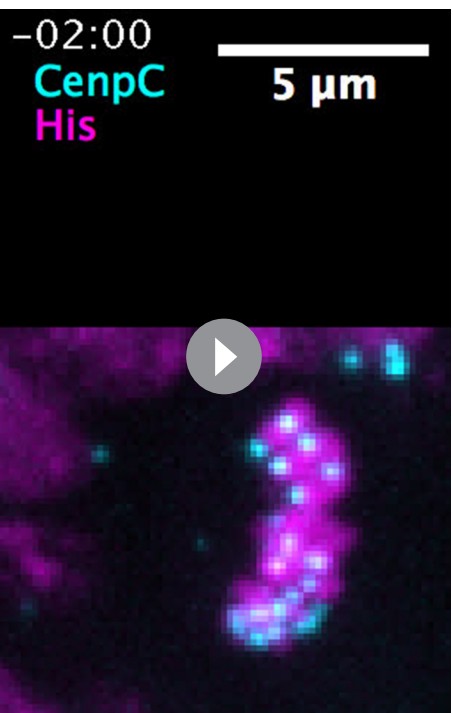

**Video 3.** This video accompanies *Figure 1—figure supplement 1E*. Live imaging of a tetraploid *HS>fzr* wing imaginal disc cell without diplochromosomes 24 hr after a 60-min heat shock to induce *fzr* overexpression with His2av-GFP in magenta to label DNA, and Cenp-C-Tomato in cyan to label kinetochores. Time Indicates minutes to the last frame of metaphase, scale bars represent 5 μm.

**Video 4.** This video accompanies *Figure 1—figure supplement 1G*. Live imaging from a tetraploid *HS>fzr* wing imaginal disc cell 24 hr after a 60-min heat shock, undergoing a tripolar anaphase with His2av-GFP in magenta to label DNA, and Cenp-C-Tomato in cyan to label kinetochores. Time Indicates minutes to the last frame of metaphase, scale bars represent 5 μm.

divisions (which lack diplochromosomes) do not exhibit obvious chromosome quartets and have a substantially reduced error rate (*Figure 1J* 4N, *Figure 1—figure supplement 1E*, *Video 3*).

We previously reported centrosome amplification to contribute to polyploid mitotic errors in *Drosophila* (*Schoenfelder et al., 2014*). However, centrosome amplification does not appear to be a major contributor to mitotic errors in the first (or subsequent) polyploid division of *HS>fzr* animals, as few tetraploid cells amplify centrosomes, and multipolar division is very rare (*Figure 1—figure supplement 1F,G*, *Video 4*). In spite of the high initial error rate caused by separation of diplochromosomes, tetraploid-aneuploid cell divisions continue to occur for at least 5 days after genome reduplication, as determined by cytology (*Figure 1H* HS+120 hr). We conclude that division of diplochromosomes in the mitotically expanding diploid progenitor tissues that we surveyed can lead to the generation of aneuploid cells, which can continue to divide (*Figure 1K*).

To determine the long-term effect of tetraploid-aneuploid divisions on tissue development, we took advantage of the fact that expression of *HS>fzr* occurs in adult progenitor tissues. We thus examined the survival of these animals to adulthood. Survival is only subtly affected in animals with mild (23.0% tetraploid [S.E.M. 4.9%]) levels of induced error-prone tetraploid progenitor division (*Figure 2A*), and resulting adult tissues appear normal (*Figure 2B*). In contrast, when tetraploidy is further increased by increasing the duration of heat shock, organism survival decreases in a tetraploid-dependent fashion (*Figure 2A*, *Figure 2—figure supplement 1A*). Together, these results show that ectopic genome reduplication in multiple progenitor tissues yields tetraploid cells with polytene metaphase diplochromosomes, which are aneuploid-prone (*Figure 1K*). These conclusions are in agreement with a previous study in the terminal embryonic division of embryos

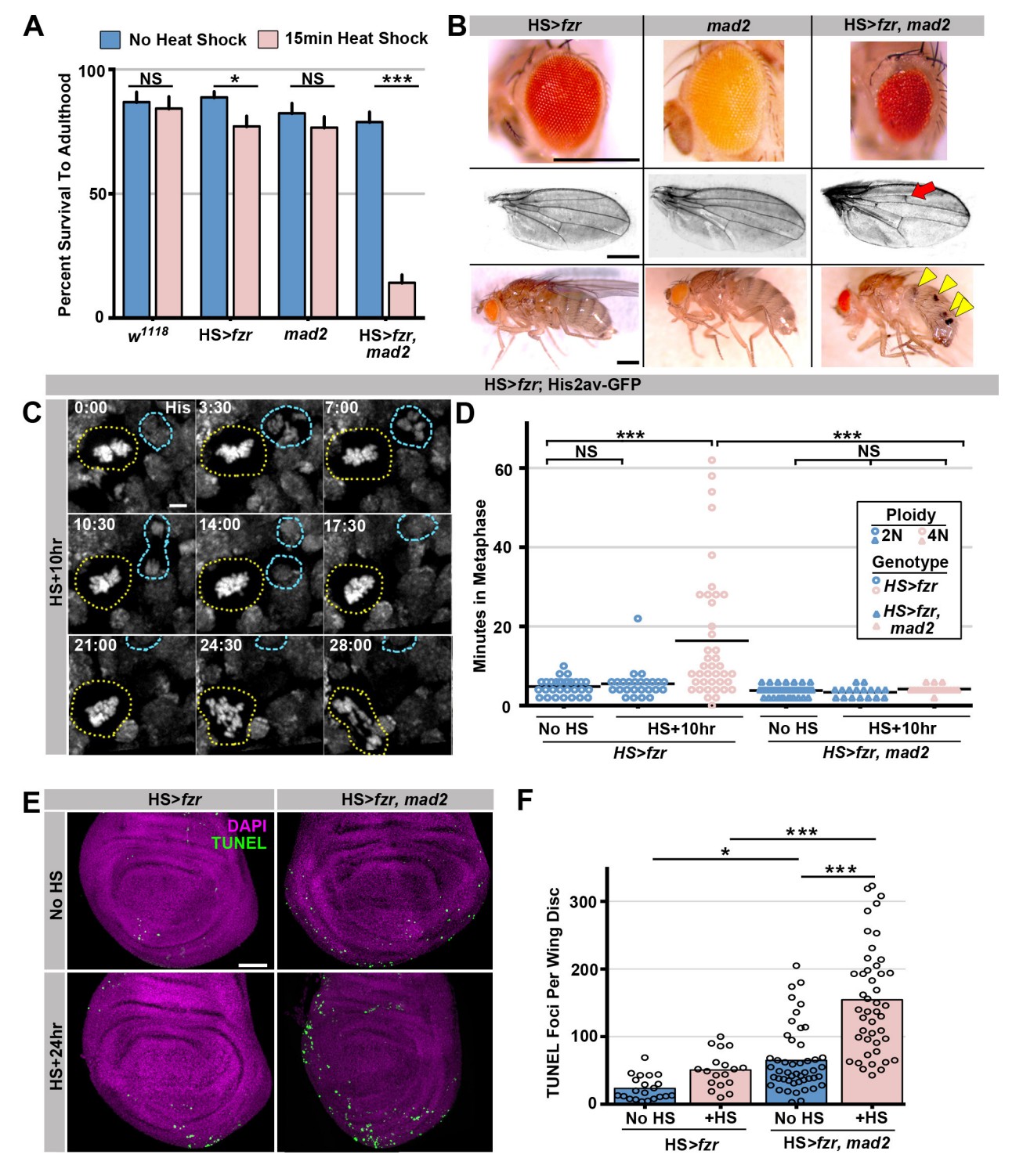

**Figure 2.** The spindle assembly checkpoint wait-anaphase response is required after ectopic genome reduplication. (A) Quantitation of survival rates from third instar larvae to adulthood of the indicated genotypes without heat shock (dark blue) or following a 15-min heat shock (light red) (which generates 23% tetraploid, see methods).Bars represent means + standard error of the mean (S.E.M) of at least 5 independent experiments, with 20 animals per experiment. *p<0.05, ***p<0.001, NS = p>0.05, t-test. (B) Representative micrographs of eyes, wings, and abdomens from *HS>fzr* alone, *mad2* alone, or *HS>fzr, mad2* flies heat shocked for 15 min as third instar larvae and then allowed to develop to adults. Red arrow indicates an extra ectopic wing vein, and yellow arrow heads indicate melanotic masses both of which are found in in *HS>fzr, mad2* adults following heat shock. (C) Representative time-lapse showing a *HS>fzr* wing imaginal disc 10 hr after a 60-min heat shock including a cell with diplochromosomes (yellow dotted

*Figure 2 continued on next page*

*Figure 2 continued*

line) and a diploid cell (blue dashed line) dividing within the same field (one of the diploid daughters drifts vertically out of the frame). The cell with diplochromosomes takes more than four times as long to enter anaphase, and division is error prone. His2av-GFP showing DNA is in white. Time indicates minutes from the start of filming. (D) The length of metaphase without *fzr* overexpression (No HS) or +10 hr after a 60-min heat shock to induce overexpression from *HS>fzr*, and *HS>fzr, mad2* larval wing imaginal disc cells. Points represent individual cell divisions, bars represent means, diploid cells are represented in dark blue, polyploid cells are represented in light red, *HS>fzr* is represented in circles, *HS>fzr, mad2* is represented in triangles. N>17 cells per column, ***p<0.001, Not Significant (NS) = p>0.05, one-way ANOVA with correction for multiple hypothesis testing. (E) Third instar larval wing imaginal discs from HS>*fzr* or HS>*fzr, mad2* stained for TUNEL in green and DAPI in magenta without heat shock (No HS) or +24 hr after a 15-min heat shock. (F) Quantification of the number of TUNEL positive foci per wing disc for *HS>fzr* and *HS>fzr, mad2* without heat shock (No HS, blue bars) or 24 hr after a 15 min heat shock (+HS, red bars). Points represent individual wing imaginal discs, bars represent mean, N $\geq$ 18 discs per condition. NS = p>0.05, * = p<0.05, *** = p<0.001, ANOVA. Scale bars represent 500 µm in B, 5 µm in C, and 50 µm in E.

The following figure supplement is available for figure 2:

**Figure supplement 1.** Supporting data regarding Mad2's role in response to diplochromosomes.

(*Vidwans et al., 2002*). We further show that such aneuploid-prone cells can continue to propagate, and that only at high frequencies are these error-prone tetraploid mitotic events lethal to the organism.

## Polyteny response 1: Spindle assembly checkpoint (SAC)-mediated anaphase delay

Our data (*Figure 1H*, *Figure 1—figure supplement 1C*) suggest that many polytene diplochromosome divisions in a variety of tissues do not lead to aneuploidy. Little is known about aneuploidy prevention mechanisms in cells with polytene chromosomes, despite the numerous mechanisms that

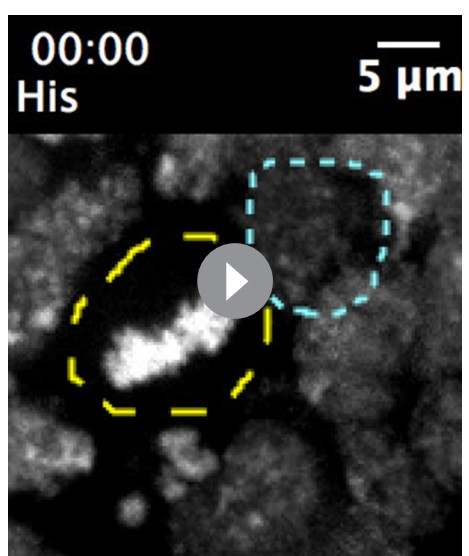

**Video 5.** This video accompanies *Figure 2C*. Live imaging of a wing disc from a *HS>fzr* animal 10 hr after a 60-min heat shock showing mitosis by a polyploid diplochromosome-containing cell (yellow dotted line) and a diploid cell (blue dotted and dashed line) in the same field, His2av-GFP labelling DNA is shown. Time Indicates minutes from the start of filming, scale bars represent 5 µm.

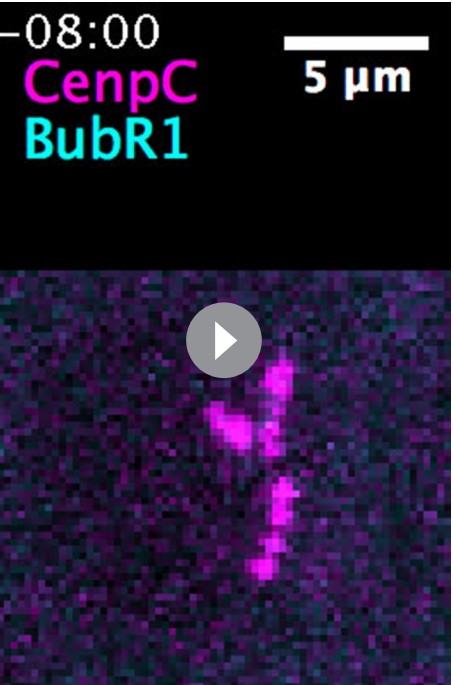

**Video 6.** This video accompanies *Figure 2—figure supplement 1C*. Live imaging of Cenp-C-Tomato in magenta to label kinetochores and BubR1-GFP in cyan during the division of a diploid cell 10 hr after a 60-min heat shock. Time indicates minutes to the last frame of metaphase. Scale bar represents 5 µm.

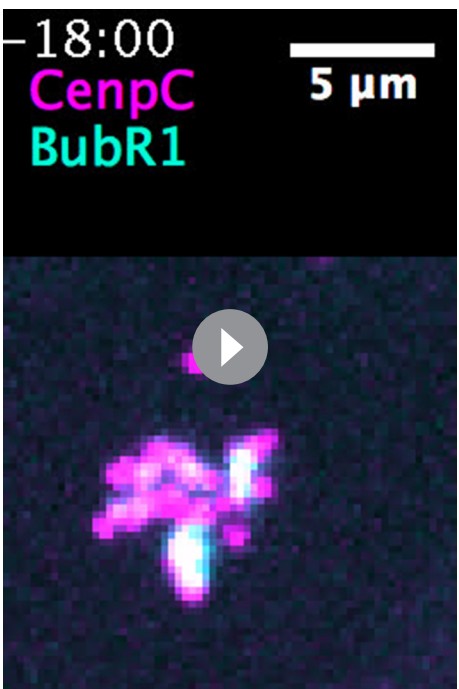

**Video 7.** This video accompanies *Figure 2—figure supplement 1C*. Live imaging showing Cenp-C-Tomato to label kinetochores in magenta and BubR1-GFP in cyan during the division of a tetraploid cell with diplochromosomes 10 hr after a 60-min heat shock. Time indicates minutes to the last frame of metaphase. Scale bar represents 5 µm.

can generate these aberrations. Through live imaging, we uncovered one such aneuploidy-prevention mechanism in cells with metaphase diplochromosomes. Because our heat shock protocol only affects cells in G2, a single *HS>fzr* pulse creates a mixed population of unaltered diploid cells and diplochromosome-containing tetraploid cells, allowing us to simultaneously live image both cell types in the same tissue. Metaphase in cells with diplochromosomes (*Figure 2C*, yellow dotted outline) is significantly longer than in diploid cells (*Figure 2C* blue dashed outline, *Video 5*, *Figure 2D*), consistent with previous work on diplochromosomes formed in *Drosophila* Securin mutants (*Pandey et al., 2005*). We thus hypothesized that diplochromosomes trigger the SAC, which activates a wait-anaphase signal until all kinetochores attach to microtubules and are under tension (*London and Biggins, 2014*; *Musacchio, 2015*). To test this model, we examined SAC-defective *mad2* null animals (*Buffin et al., 2007*; *Emre et al., 2011*) *Figure 2—figure supplement 1B*). Using live imaging of wing imaginal discs before and after heat shock, we find that loss of *mad2* eliminates the lengthened period of metaphase caused by diplochromosomes (*Figure 2D*). When the checkpoint is active unattached or misattached kinetochores generate a wait-anaphase signal by localizing SAC proteins such as BubR1 to those kinetochores (*Musacchio, 2015*). To confirm that diplochromosomes have localized SAC proteins we co-imaged kinetochores and BubR1 in wing disc cells after *fzr* expression (*Royou et al., 2010*). We find that in diploid cells BubR1-GFP is clearly evident on kinetochores following nuclear envelope break down and remains there until anaphase. This signal is relatively evenly spread across all the kinetochores (*Figure 2—figure supplement 1C*, diploid, *Video 6*). In contrast BubR1-GFP remains localized for much longer in cells with diplochromosomes and is often localized strongly to a specific kinetochore group rather than evenly distributed, suggesting that a subset of diplochromosomes may have trouble forming attachments that satisfy the checkpoint (*Figure 2—figure supplement 1C*, diplochromosomes, *Video 7*) From these data, we conclude that diplochromosomes trigger a SAC wait-anaphase response.

Although important for mitosis in cultured *Drosophila* S2 cells (*Orr et al., 2007*), the Mad2-directed SAC is reported to be dispensable in *Drosophila* tissue mitosis (*Buffin et al., 2007*; *Emre et al., 2011*). *mad2* null animals are viable with no obvious tissue defects (*Figure 2B*; *Buffin et al., 2007*) due in part to an apoptotic response (*Morais da Silva et al., 2013*). In contrast, the Mad2-dependent wait-anaphase response is essential during development of *HS>fzr* animals. Even at low levels of tetraploidy, which affect the survival of *HS>fzr* animals only slightly, few *HS>fzr, mad2* animals survive to adulthood (15.4%, *Figure 2A*, *Figure 2—figure supplement 1A*). To understand why *HS>fzr, mad2* animals have survival defects, we analyzed the surviving animals. In these animals, we find a variety of developmental defects in normally diploid tissues, including smaller eyes, ectopic wing veins, and melanotic abdominal masses (*Figure 2B*). Increased apoptosis is associated with these tissue malformation phenotypes, as progenitor tissue from *HS>fzr, mad2* animals have much higher rates of apoptotic cell death as shown by both TUNEL labeling (*Figure 2E,F*), and cleaved caspase staining (*Figure 2—figure supplement 1D,E*). Further, 100% of *mad2* diplochromosome divisions exhibit lagging chromosomes, or DNA bridges (*Figure 2—figure supplement 1F*, *Video 8*), compared with 80% of divisions in *HS>fzr* cells. These data may suggest that

diplochromosomes are likely susceptible to at least two classes of mitotic errors, one that can be corrected by the SAC, and one that cannot. It is also likely that the *mad2* diplochromosome divisions are qualitatively more erroneous than in WT, which may account for differences in survival and tissue phenotype between these genotypes. Taken together, our data identify an important role for the Mad2-dependent SAC in delaying anaphase in the presence of metaphase polytene diplochromosomes.

## Polyteny response 2: Separation Into Recent Sisters (SIRS)

Having defined a response to mitosis after ectopic genome reduplication, we next asked if this same mechanism operates in a tissue that we previously found to divide after programmed genome reduplication. In earlier work we found *Drosophila* rectal papillar cells (hereafter: papillar cells) naturally undergo two *fzr*-dependent endocycles during the 2nd larval instar to generate octoploid cells and then divide, on average, two times during pupal development. An intervening S-phase accompanies these polyploid divisions, and cells at the papillar base undergo one additional S-phase after the final polyploid mitosis (*Figure 3A*; *Fox et al., 2010*; *Schoenfelder et al., 2014*). Thus, as with *HS>fzr* induction in diploid tissues, papillar development naturally involves genome reduplication followed by mitosis. Our previous work established that papillar mitoses can be error prone, so the same problems with dissociating polytene chromosomes in *HS>fzr* tissues could be responsible for a portion of these errors during papillar divisions. However, we previously did not see, in hundreds of observed cells, any instances of metaphases with persistent polyteny in papillar cells, suggest-

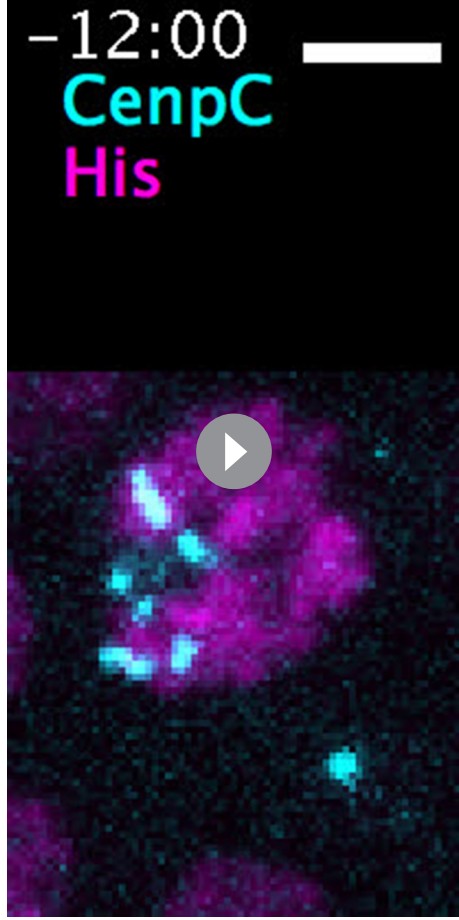

**Video 8.** This video accompanies *Figure 2—figure supplement 1F*. Live imaging showing His2av-GFP in magenta and Cenp-C-Tomato in cyan during the division of a *HS>fzr,mad2* tetraploid cell with diplochromosomes 10 hr after a 60-min heat shock. Time indicates minutes from the last frame of metaphase. Scale bar represents 5 μm.

ing that papillar cells somehow avoid mitosis of polytene chromosomes. Through careful re-examination of the first octoploid metaphase, we confirmed that papillar chromosomes in these octoploid cells are arranged in individual sister chromatid pairs (*Figure 3B''* inset, C). This suggested two possibilities: 1) papillar cells never form polytenes, or 2) papillar cells form polytenes, but somehow separate into recent sister pairs prior to the first metaphase.

To distinguish these two possibilities, we examined papillar karyotypes from the moment chromosome condensation could be detected. Papillar cells re-enter mitosis from a G2-like state, as evidenced by expression of the G2/M regulator Cdc25/string just before the onset of pupal cell cycles (*Fox et al., 2010*) (*Figure 3—figure supplement 1A*). At time points early in the first mitosis, we indeed find that papillar chromosomes are polytene (*Figure 3B*, Polytene). In these polytenes, we again see examples of cells where the centromeric regions are no longer tightly associated, as we did in our studies of *HS>fzr*-induced polyteny. However, unlike in cells with induced polyteny, the centromeres in papillar polytene cells are able to not only separate into groups of homologs, but to further separate into individual sister chromatid pairs (asterisks in *Figure 3B* and *Figure 3—figure supplement 1B* vs. *Figure 1F*, also see discussion). In flies heterozygous for inversion-containing balancer chromosomes, papillar polytene structure is locally perturbed, likely due to the disruption of somatic homolog pairing (*Figure 3—figure supplement 1B*). At this early mitotic time point, we

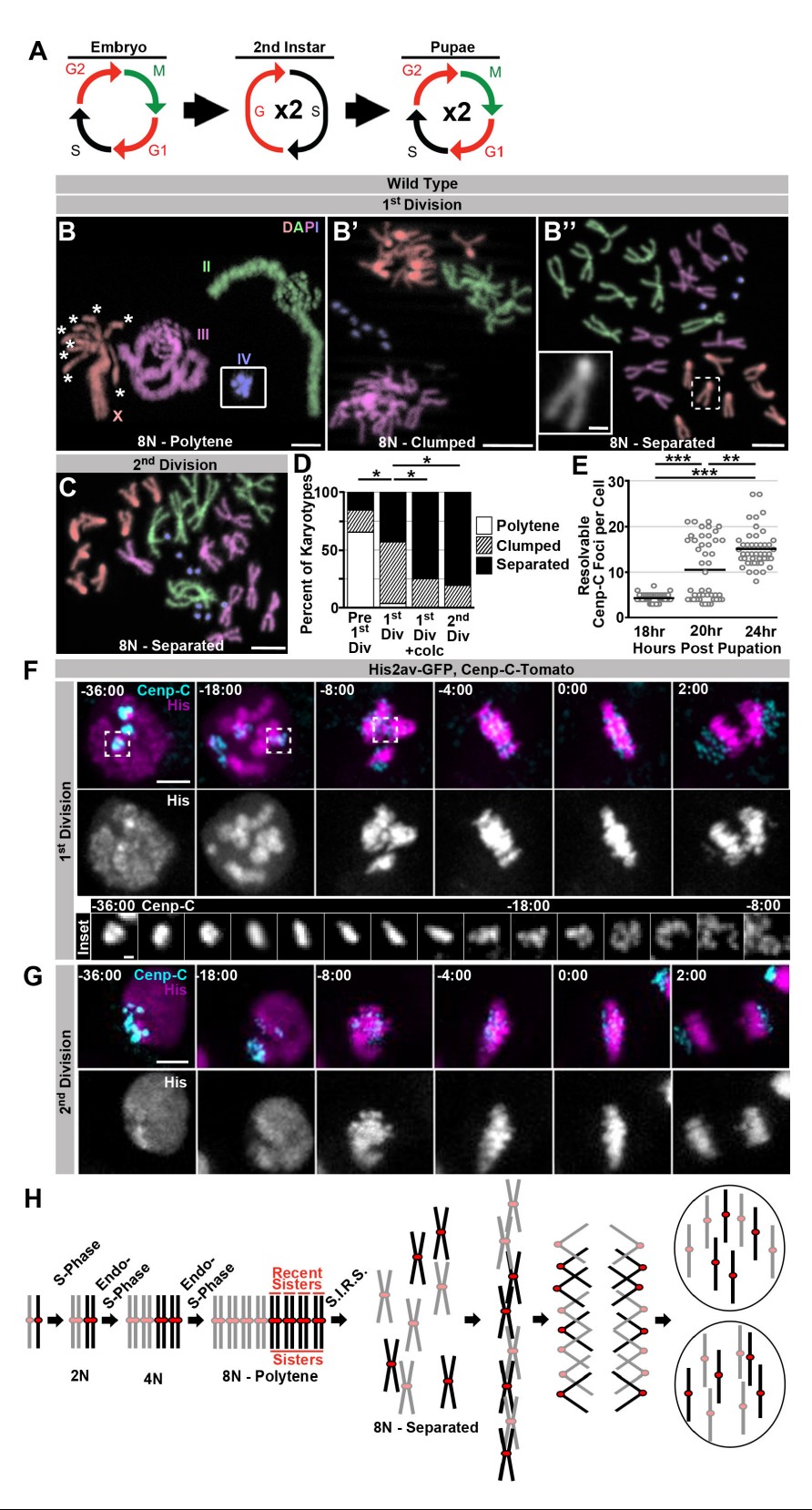

**Figure 3.** Programed genome reduplication in papillar cells is followed by Separation Into Recent Sisters (SIRS), which individualizes polytene chromosomes into recent sister pairs. (**A**) A model of the cell cycles in *Drosophila* papillar cells. These cells undergo two rounds of the endocycle in the
*Figure 3 continued on next page*

*Figure 3 continued*

2nd instar to reach 8N/16C, then enter a G2-like state, then undergo, on average, two cell divisions with intervening S-phases during pupation (*Fox et al., 2010*; *Schoenfelder et al., 2014*). (B) Karyotypes of papillar cells during the 1st polyploid division, (B–B''). Chromosomes are pseudocolored according to type and labeled in panel B. Panel B inset shows the 4th chromosomes, which were out of frame. When chromosomes first condense following genome reduplication, they are in a polytene configuration (asterisks indicate the 8 separated centromere pairs of an otherwise polytene X chromosome). This cell contains a heterozygous pericentric inversion on the third chromosome caused by the presence of a balancer chromsome. B' Example of the clumped configuration in early mitosis of the first papillar division. B'' Example of fully separated 1st division papillar chromosomes. No diplochromosomes are present (compare X chromosome in inset to inset in *Figure 1F'*). Note- one second chromosome contains a DNA break, which are common in wild type papillar cells (*Fox et al., 2010*; *Bretscher and Fox, 2016*). (C) Karotype of papillar chromosomes during the 2nd polyploid division. Chromosomes are pseudocolored according to type as in *Figure 3B*. At the second division almost all cells show chromosomes fully separated into sister pairs. (D) Percentage of cells with polytene chromosomes, recent sisters clumped, or recent sisters clearly separated from four time points: prior to the first division (following treatment with Calyculin A to visualize pre-mitotic chromosome structure- see Materials and methods), during the first division (no drug treatment), during the first division (following treatment for 30 min with colcemid to enrich for late metaphase (1st Div + colc)), and during the second division (no drug treatment). *p<0.05 compared to 1st Division, chi-squared test, N $\geq$ 26 karyotypes per treatment from at least 5 animals. (E) Quantification of the number of resolvable Cenp-C-Tomato foci in fixed papillar cells during the course of pupation (expressed in hours post pupation). Before the first mitosis (18 hr) each cell has an average of 4.1 kinetochore foci closely corresponding to the haploid chromosome number, following the first division (24 hr) cells average 15.1 foci per cell. At 20 hr some cells have divided and others are yet to divide and the distribution is bimodal. Circles represent individual cells. Bars represent the mean of 3 animals per time point and 15 cells per animal. ***p<0.001, **p<0.01, by Kruskal-Wallis one-way ANOVA. (F) Live imaging of the 1st divisions from wild type papillar cells shows the SIRS process. Cenp-C-Tomato (Cenp-C) is in cyan, His2av-GFP (His) is in magenta. Time represents minutes to the last frame prior to anaphase. In the 1st division kinetochores from a group of homologs are tightly clustered prior to division. At -18:00 min. relative to anaphase, chromosome condensation has begun and polytene chromosomes are visible (See His channel). Dispersal continues until individual pairs of sister kinetochores are evident at metaphase. The inset shows the Cenp-C-Tomato channel of a single kinetochore focus from time frames -36 min to -8 min. (G) Live imaging of the 2nd division from a wild type papillar cell. In contrast to the first division many discrete kinetochore foci are evident at time-points prior to the onset of mitosis, and polytene chromosomes are never evident. (H) A model for a pair of homologs undergoing 2 rounds of endo-S-phase to become a polytene 16C chromosome. The polytene chromosome then separates into pairs composed of only the most recent sister chromatids during mitosis, and each sister then segregates to opposite poles at anaphase. Scale bar represents 5 μm, except in insets in B'' and F where it represents 1 μm.

The following figure supplement is available for figure 3:

**Figure supplement 1.** Supporting data regarding SIRS.

---

also observe cells where polytenes are absent. Instead, in these cells, sister chromatid pairs and homologs of each chromosome type are separated but remain clumped closely together, as if the polytene chromosome recently separated into pairs containing only the most recent sister chromatids (*Figure 3B', D*, Clumped). Neither the polytene nor the clumped configurations remain during the second division (*Figure 3C,D*), suggesting a specific chromosome structure is present early in the first division of papillar cells. Similarly, by Fluorescent In Situ Hybridization (FISH) we find examples of both closely associated (polytene) and dispersed (separated/non-polytene) signals during the period of the first papillar mitosis (*Figure 3—figure supplement 1C,C'*). Thus, a key difference between the response to genome reduplication between papillar and *HS>fzr* cells is the elimination of polyteny before anaphase in papillar cells.

To examine if a majority of (if not all) papillar cells transition from polytene to separate/non-polytene chromosomes during the first mitosis, we used drug treatment to isolate specific chromosome structures during the transition into the first papillar division. To enrich for early mitotic and pre-mitotic chromosomes, we induced Premature Chromosome Compaction (PCC) in papillar cells at a time point just prior to the first mitosis (Methods). PCC causes interphase chromosomes to condense and makes it possible to visualize interphase chromosome structure by standard cytological methods (*Figure 3—figure supplement 1B*). Using this technique, we find that in pre-mitotic papillar tissue, clear polytene chromosomes are present in nearly every cell (*Figure 3D* Pre 1st Div, *Figure 3—figure supplement 1B*). If we instead enrich for cells in metaphase of the first mitosis by treating with the spindle poison colcemid, we find zero examples where chromosomes are still polytene. In these metaphase-enriched samples, all chromosomes are separated into recent-sister pairs, and even cells with clumped chromosomes are rare (*Figure 3D*, 1st Div +colc). Thus, our pharmacological studies further suggest that essentially all genome-reduplicated papillar cells are programmed to completely eliminate polytene chromosomes as cells progress into the first metaphase.

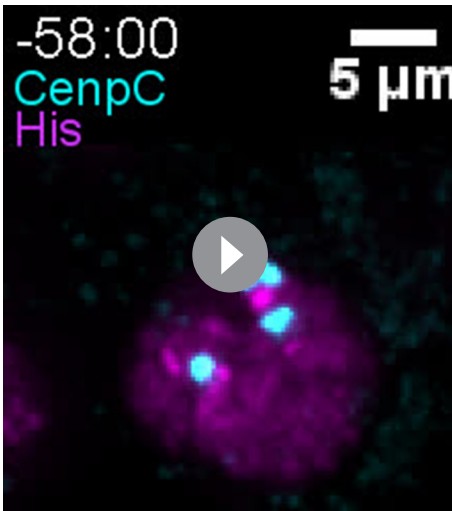

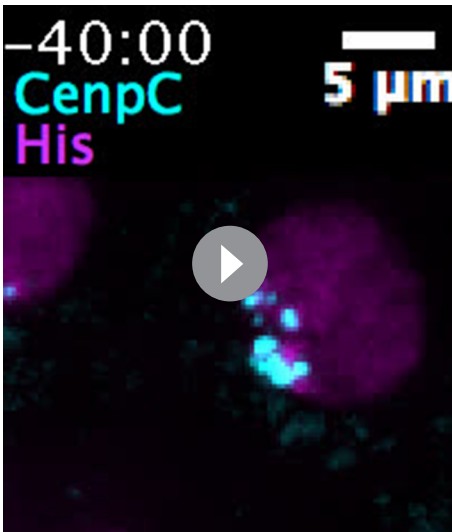

**Video 9.** This video accompanies *Figure 3F*. Live imaging of a papillar cell undergoing a first division, including the SIRS process showing His2av-GFP in magenta, and Cenp-C-Tomato in cyan. SIRS is most evident at the centromere which transitions from four tight foci prior to mitosis to many foci at anaphase. Polytene chromosomes are visible 18:00 min before anaphase. Time Indicates minutes to the last frame of metaphase, scale bars represent 5 μm.

**Video 10.** This video accompanies *Figure 3G*. Live imaging showing His2av-GFP in magenta, and Cenp-C-Tomato in cyan from a papillar cell undergoing a second division. Time Indicates minutes to the last frame of metaphase, scale bars represent 5 μm.

To observe the temporal dynamics of the pre-anaphase elimination of papillar polyteny, we used live imaging, using the same markers used to image diplochromosome division. Prior to the first papillar mitosis, the kinetochores from each homolog are closely associated into an average of 4.1 large foci, close to the haploid number of distinct chromosomes (4 for females and 5 for males due to X/Y un-pairing, *Figure 3E*). As time progresses in the first division, it is possible to watch these large kinetochore foci disperse into many smaller foci prior to metaphase (*Figure 3F*, inset, *Video 9*). In contrast, prior to the second division kinetochores are already separated into many more foci (an average of 15.1 observably distinct foci per cell) before entry into mitosis (*Figure 3E*). During the second division, the number of resolvable foci remains essentially constant (*Figure 3G*, *Video 10*). Additionally, the histone marker reveals that polytene chromosomes are visible when chromosomes first condense and can then be seen to disperse during the first but not the second division (*Figure 3F v. 3G*, -18:00 min). This result confirms the model that genome-reduplicated papillar cells eliminate polyteny during the first mitosis, then undergo an intervening S-phase before the next division (*Figure 3A*). We also confirmed that each clump of 4 or 5 pre-first division centromeres only contains a single chromosome type. To do so, we took advantage of the fact that dosage compensation in flies relies on upregulation of transcription on the male X chromosome via the Dosage Compensation Complex (*Conrad and Akhtar, 2011*). By live imaging papillar cells expressing the DCC complex protein MSL3 tagged with GFP, which localizes only to the male X-chromosome (*Strukov et al., 2011*), we find that indeed only a single Cenp-C-Tomato focus is MSL3-GFP positive prior to polytene dissociation (*Figure 3—figure supplement 1D*, *Video 11*). Taken together, we find papillar cells avoid mitosis of polytene chromosomes in part by undergoing a pre-anaphase chromosome separation process we term Separation Into Recent Sisters (SIRS, *Figure 3H*).

Previously, we confirmed that a similar polyploid mitotic program occurs in the developing hindgut of the mosquito *Culex pipiens* (*Fox et al., 2010*). Interestingly, classical descriptions of mitosis in this part of the *Culex* hindgut seem to suggest a polytene organization is present only early in the first polyploid mitosis (*Grell, 1946*). In agreement with this observations, we find Phospho-Histone H3 positive polytene chromosomes during the period of the first polyploid mitosis (*Figure 3—figure supplement 1E*, *Video 12*). From our *Drosophila* and *Culex* studies, we conclude that unlike cells

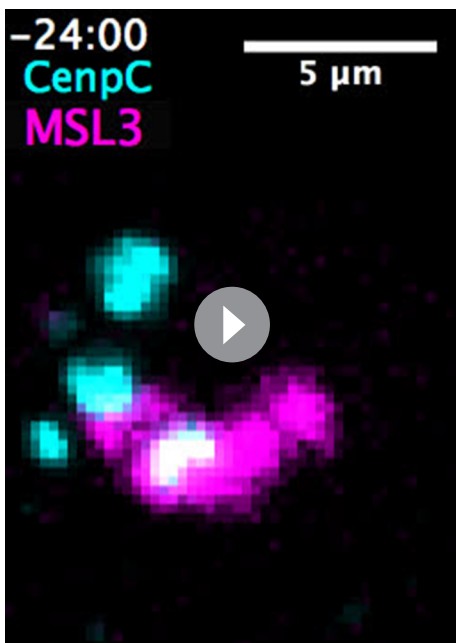

**Video 11.** This video accompanies *Figure 3—figure supplement 1D*. Live-imaging showing MSL3-GFP in magenta and Cenp-C-Tomato in cyan from a male papillar cell undergoing a first division. Only a single Cenp-C-Tomato foci is MSL-3-GFP positive. Time Indicates minutes to the last frame of metaphase, scale bars represent 5 µm.

that enter metaphase with polytene chromosomes, a separate mechanism, SIRS, can eliminate polyteny as cells enter metaphase.

## Spindle-independent mitotic timing by Mad2 promotes efficient SIRS

Our dual genome-reduplication systems identified two distinct cellular responses to polytene chromosomes: a Mad2-dependent response that delays anaphase when polytenes remain at metaphase, and a SIRS response that eliminates polyteny as cells enter metaphase. Despite the lack of metaphase polytene chromosomes in papillar cells, we also identified an important role for Mad2 during SIRS. Because *mad2* loss has no reported mitotic defects in *Drosophila* animals (*Buffin et al., 2007*), we were surprised to find that first division *mad2* papillar cells exhibit a substantial increase in DNA bridges (*Figure 4A,B*, *Video 13*, *Video 14*). We did not detect similar defects during papillar mitosis of animals null for *mad1,* another SAC component (*Figure 4A,B*, *Video 15*). Thus, a Mad1 independent function of Mad2 is important in cells during SIRS.

In *Drosophila*, Mad2 plays a conserved, cell type-dependent role in regulating NEBD-to-anaphase onset timing (*Buffin et al., 2007*; *Meraldi et al., 2004*; *Rodriguez-Bravo et al., 2014*; *Yuan and O'Farrell, 2015*). As for the wait-anaphase response, this mitotic timing role involves Mad2 inhibition of the Anaphase Promoting Complex. However, Mad2's control of overall mitotic timing is irrespective of SAC kinetochore attachment surveillance (*Meraldi et al., 2004*; *Rodriguez-Bravo et al., 2014*). Interestingly, *Drosophila* Mad1 is reported to be dispensable for regulation of NEBD-to-anaphase timing (*Emre et al., 2011*). Given the lack of *mad1* phenotypes with respect to the first papillar division, we thus hypothesized that SIRS enables papillar cells to bypass the SAC-mediated anaphase delay, and that Mad2 control of overall mitotic timing is important during SIRS. If so, one would predict cells undergoing SIRS to not trigger an anaphase delay, but to still depend on mitotic timing.

To first test if papillar cells employ the SAC wait-anaphase in response to polytene chromosomes, we treated animals with colcemid, a known SAC wait-anaphase trigger. This treatment increases the mitotic index of wild type papillar cells, whereas the mitotic index of *mad2* null animals is unaffected (*Figure 4C,D*) Thus, spindle defects trigger the SAC wait-anaphase response in papillar cells. We next asked if the SAC wait-anaphase responds to polytene chromosomes during SIRS. If so, the first divisions (polytenes present) should have a longer metaphase than the second division (polytenes absent). However we find that metaphase is not any longer in the first papillar division than in the second papillar division, while in contrast metaphase is almost twice as long in wing cells with diplochromosomes than in those that are polyploid but lack

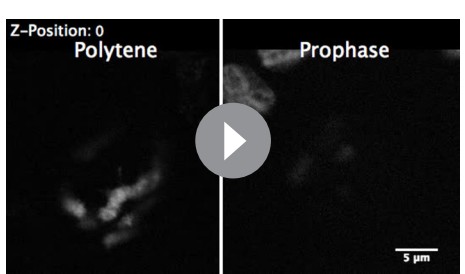

**Video 12.** This video accompanies *Figure 3—figure supplement 1E*. Video showing sequential z-planes from a fixed ileum of *Culex* pipiens with mitotic cells that are pre-SIRS (left) and post-SIRS (right) stained with Phospho-Histone H3. scale bars represent 5 µm.

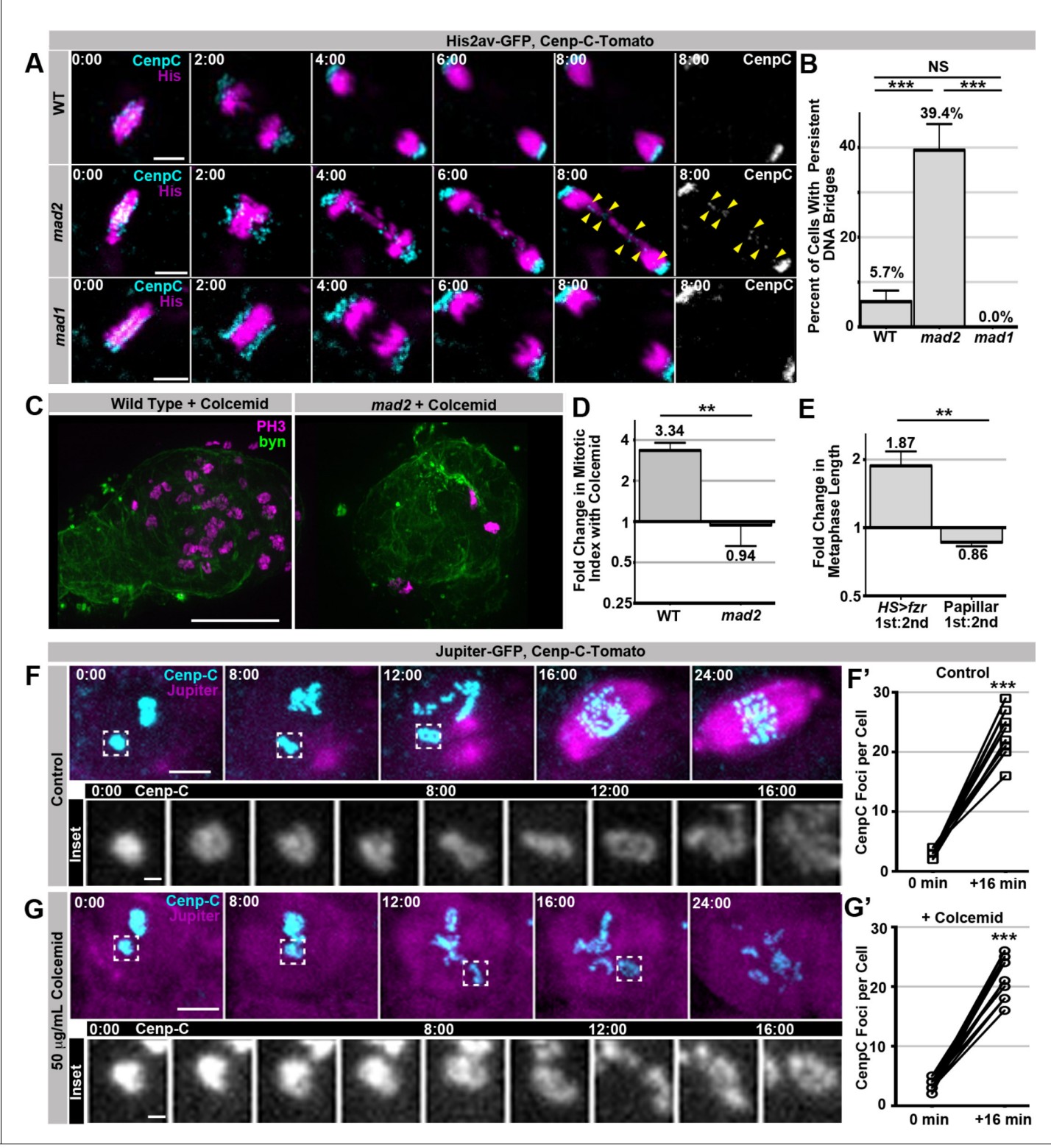

**Figure 4.** SIRS does not depend on the SAC wait-anaphase response or formation of a mitotic spindle. (A) Representative micrographs of wild type (WT), *mad2*, and *mad1* cell 1st divisions beginning in the last frame of metaphase (0:00) and continuing through eight minutes of anaphase.Cenp-C-Tomato showing kinetochores in cyan, His2av-GFP showing DNA in magenta. Yellow arrowheads show kinetochores that are part of a bridge between the two poles in a *mad2* cell. Time represents minutes from the last frame prior to anaphase. (B) Quantification of the frequency of persistent DNA bridging observed 4 min after the onset of the 1st division anaphase from papillar cells in wild type (WT), *mad2*, and *mad1* animals. Bars represent the

*Figure 4 continued on next page*

*Figure 4 continued*

mean of all cell divisions, + Standard Error of the Mean (S.E.M.) ***p<0.001, t-test. (C) Representative images of a single pupal rectums from wild type or *mad2* animals treated with colcemid for 60 min prior to fixation and stained for Phospho-Histone H3 (PH3) positive nuclei in magenta and expressing GFP under the control of *brachyenteron (byn,* a hindgut marker) in green. (D) The fold increase in the number of polyploid mitotic cells per hindgut from wild type and *mad2* animals following treatment with colcemid compared to without colcemid. A value of 1 equals no difference. Bars represent mean fold change (+ SEM), and are labeled with the mean value. **p<0.01, t-test, N $\geq$ 8 animals per condition. (E) The fold increase in metaphase length for *HS>fzr* polyploid cells with (1st) and without polytene diplochromosomes (2nd) compared to papillar cells with (1st division) and without (2nd division) polytene chromosomes. A value of one equals no difference between 1st and 2nd divisions. The increase in *HS>fzr* wing cells indicates that metaphase polytene diplochromosomes trigger the spindle assembly checkpoint, but papillar polytene chromosomes do not. Bars represent means (+S.E.M.), and are labeled with the mean value. *p<0.05, t-test, N $\geq$ 22 cells per condition from at least 5 animals. (F) Live imaging of a cell expressing Cenp-C-Tomato in cyan and Jupiter-GFP in magenta undergoing SIRS in the presence of a vehicle control. Time represent minutes before the onset of SIRS. Inset shows the dispersal of a single Cenp-C-Tomato kinetochore at all the time points between 0 min and 16 min. (F') shows the number of resolvable Cenp-C-Tomato foci from prior to SIRS (0 min) and after SIRS (16 min), points represent individual cells with the two time points connected by a line. ***p<0.001, t-test N = 12 divisions from 2 animals. (G) Live imaging of a cell expressing Cenp-C-Tomato in cyan and Jupiter-GFP in magenta undergoing SIRS in the presence of a colcemid. Time represent minutes from the onset of SIRS. Inset shows the dispersal of a single Cenp-C-Tomato kinetochore at all the time points between 0min and 16 min. (G') shows the number of resolvable Cenp-C-Tomato foci from prior to SIRS (0 min) and after SIRS (16 min), points represent individual cells with the two time points connected by a line. ***p<0.001, t-test, N = 15 cells from 5 animals. Scale bar represents 5 μm except in insets of F, and G where they represent 1 μm.

diplochromosomes (*Figure 4E*). We then tested if triggering the SAC wait-anaphase response can prevent or delay SIRS completion. We find SIRS occurs on schedule even in the presence of colcemid concentrations that are sufficient to eliminate a detectable spindle and inhibit anaphase (*Figure 4F, G*, *Video 16*, *Video 17*). We conclude that: a) SIRS is not regulated by the SAC wait-anaphase response, and b) chromosome separation during SIRS does not require a mitotic spindle.

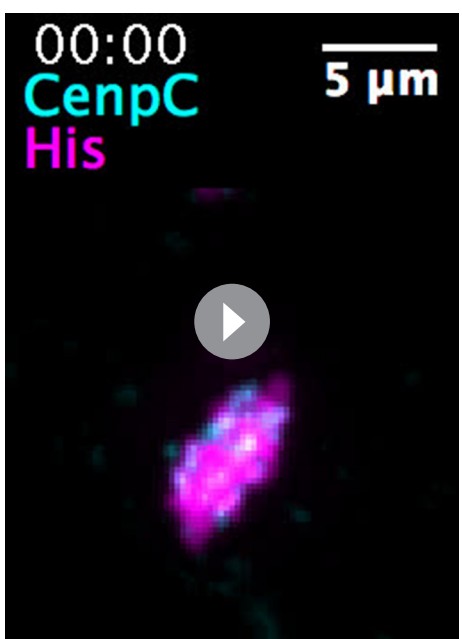

**Video 13.** This video accompanies *Figure 4A*. Live imaging showing His2av-GFP in magenta, and Cenp-C-Tomato in cyan from a wild type papillar cell during anaphase of the first mitosis. Minutes indicates time before the onset of anaphase. No DNA bridge is present.

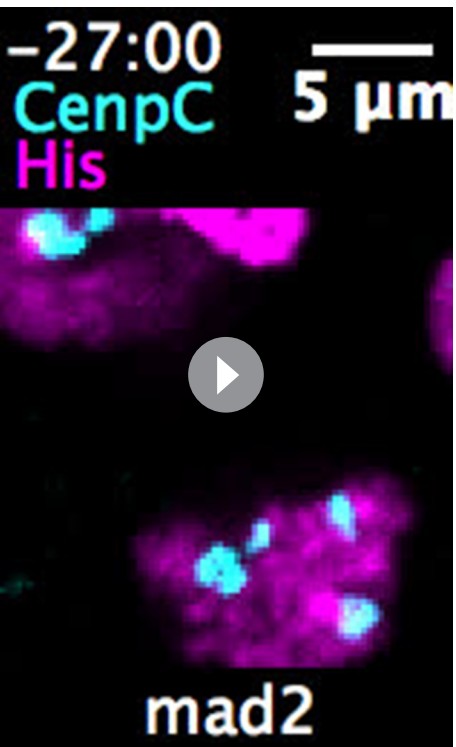

**Video 14.** This video accompanies *Figure 4A*. Live imaging showing His2av-GFP in magenta, and Cenp-C-Tomato in cyan from a *mad2* papillar cell from DNA condensation through anaphase of the first mitosis, including formation of a DNA bridge. Minutes indicates the time to before the onset of anaphase.

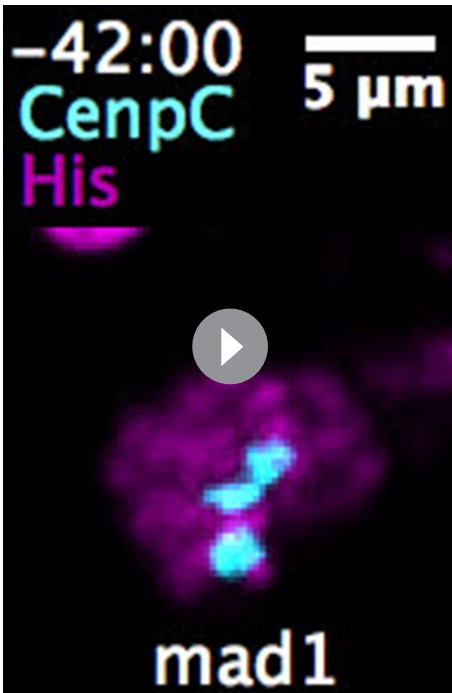

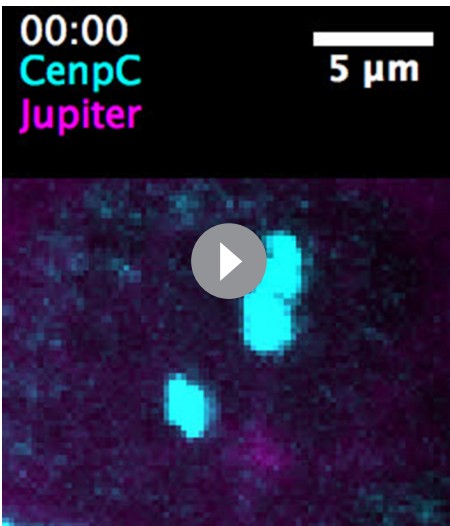

**Video 15.** This video accompanies *Figure 4A*. Live imaging showing His2av-GFP in magenta, and Cenp-C-Tomato in cyan from a *mad1* papillar cell during from DNA condensation through anaphase of the first mitosis. No DNA bridging is present in *mad1* papillar cells. Time indicates minutes from the onset of anaphase.

**Video 16.** This video accompanies *Figure 4F*. Live imaging showing Jupiter-GFP in magenta and Cenp-C-Tomato in cyan from a first division papillar cell undergoing SIRS in control imaging media. Time indicates minutes from the start of filming.

We next tested whether Mad2-dependent control of overall mitotic timing is crucial for efficient SIRS. Using an NEBD marker, we first confirmed that Mad2 regulates NEBD-to-anaphase timing and that *mad2* cells spend significantly less time in mitosis than wild type cells (*Figure 5A,B*, *Video 18*, *Video 19*). We also find that Nuclear Envelope Breakdown and the onset of SIRS are synchronous in wild type cells and that SIRS generally continues until up to the onset of anaphase (*Figure 5C*). This suggests that the rapid mitosis in *mad2* cells might lead to a failure of complete SIRS, which could cause the resulting DNA bridges. In our live imaging, we saw evidence that a pre-SIRS group of homologs would often fail to completely disperse prior to anaphase (*Figure 5A*, *mad2*, yellow arrowhead). To quantify this, we generated heat maps and line profiles of centromere signals at the metaphase plate. We performed this analysis just prior to the onset of mitosis in wild type cells, *mad2* cells that did not generate bridges, and *mad2* cells that did generate DNA bridges (*Figure 5D*). From these measurements, we found that SIRS fails to complete before anaphase in *mad2*

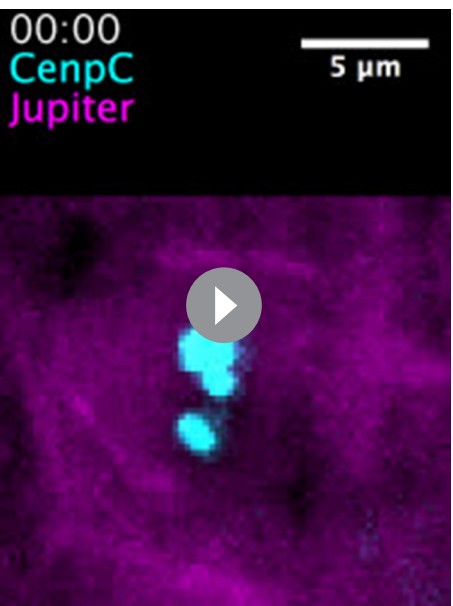

**Video 17.** This video accompanies *Figure 4G*. Live imaging showing Jupiter-GFP in magenta and Cenp-C-Tomato in cyan from a first division papillar cell undergoing SIRS in the presence of a colcemid. Colcemid prevents spindle formation so the Jupiter remains diffuse. Time indicates minutes from the onset of filming.

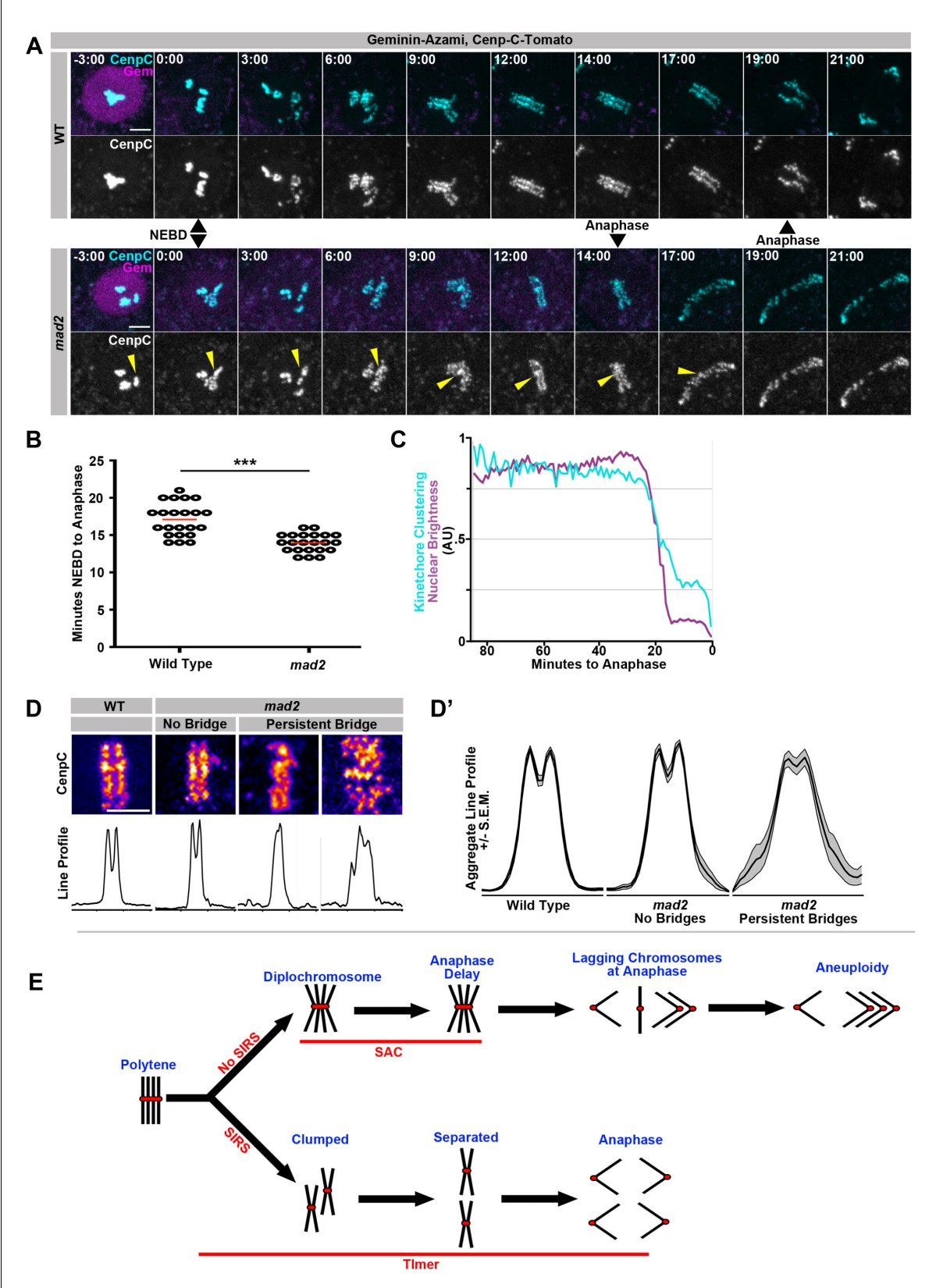

**Figure 5.** SIRS is dependent on Mad2-dependent mitotic timing. (**A**) Live imaging of a representative wild type and *mad2* cell expressing geminin-Azami (magenta) in and Cenp-C-Tomato (cyan) during the 1ˢᵗ papillar division. Just the Cenp-C-Tomato channel is also shown. Nuclear Envelope

*Figure 5 continued*

Breakdown (NEBD) can be seen when the geminin signal goes from nuclear to cytoplasmic. Time represents minutes from NEBD. The *mad2* cell reaches anaphase more quickly after NEBD than the wildtype cell (14 min to 19 min). Yellow arrows indicate a Cenp-C foci in *mad2* that appears to fail SIRS and is still partially clumped at anaphase. (B) The length of time from NEBD to anaphase in wild type and *mad2* papillar cells. Points represent individual cell divisions, red bar represents mean (17.1 min for wild type, 14.0 min for *mad2*). ***$p<0.001$, t-test, N = 22 cell divisions for each condition from at least 5 animals. (C) Quantification of the intensity of geminin-Azami in magenta and a measure of kinetochore clusteredness in cyan over time from wild type cell. 0 min represents the onset of anaphase. Both measures decline synchronously at the onset of NEBD. Data represents the mean of 22 cells. (D) Representative images of Cenp-C-Tomato forming the metaphase plate of WT or *mad2* cells immediately prior to the onset of anaphase with reds indicating more Cenp-C-Tomato signal and blue indicating less signal (Top) and line graphs measuring the total signal intensity from left to right (Bottom), in call cases the eventual division is in the same left-right orientation. *mad2* metaphases were split into those that did not generate a persistent DNA bridge at anaphase (no bridge) and those that did (persistent bridge). (D') Aggregate plots of the line graph and the confidence interval for each category. *mad2* cells that formed bridgs are significantly more variable than wild type or *mad2* without bridging. N > 13 cells per category. (E) Model: A simplified model in which four sisters from a single round of genome reduplication are shown. In cells with SIRS (down arrow) polytenes fully split into individual sister pairs and with a functioning mitotic timer complete SIRS and divide evenly. However, in the absence of the timer anaphase is precocious and DNA bridges result from incompletely resolved polytene chromosomes. In cells without SIRS (upper arrow), diplochromosomes result. The spindle assembly checkpoint (SAC) delays cells in metaphase and reduces but does not eliminate aneuploidy during the ensuing anaphase. In the absence of a checkpoint, cell death results from errant diplochromosome divisions.

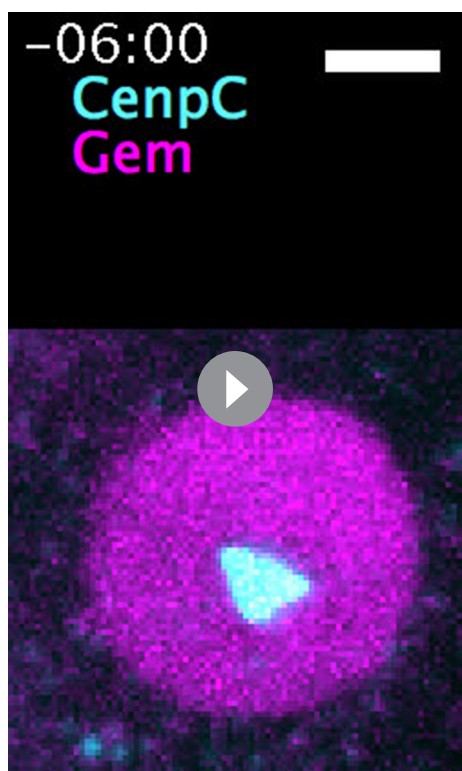

**Video 18.** This video accompanies *Figure 5A*. Live imaging showing Geminin-Azami in magenta, and Cenp-C-Tomato in cyan in a wild type papillar cell from prior to Nuclear Envelope Breakdown through anaphase. The geminin signal is nuclear before the onset of mitosis. Concurrent with NEBD is the onset of SIRS. Anaphase takes place 19 min after NEBD. Time indicates minutes from the first frame after NEBD. Scale bar represents 5 μm

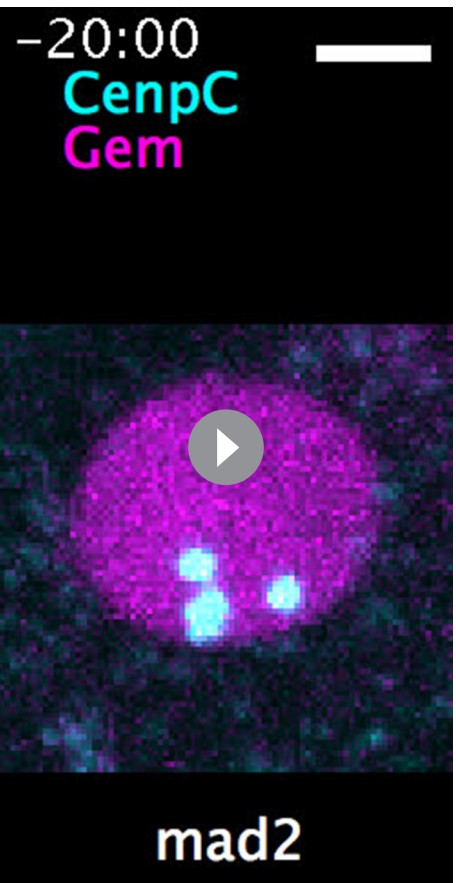

**Video 19.** This video accompanies *Figure 5A*. Live imaging showing Cenp-C-Tomato in cyan and Geminin-Azami in magenta in a *mad2* papillar cell from prior to NEBD through anaphase. Anaphase is 14 min after NEBD and Cenp-C-tomato clumps are still evident. Time indicates minutes from the first frame after NEBD. Scale bar represents 5 μm

animals, leading to a high variance of centromere intensity signal across the metaphase plate (reflecting failure of centromere dissociation/SIRS completion). This high variance disrupts the bilateral symmetry of the metaphase plate in *mad2* cells that form DNA bridges (*Figure 5D'*). Taken together, our data show mitotic fidelity after genome reduplication is improved by one of two Mad2-dependent functions: 1) in the presence of metaphase polytene chromosomes a Mad2-dependent wait-anaphase signal is generated, 2) the efficient elimination of polytenes as cells enter metaphase by SIRS requires a Mad2 (but SAC wait-anaphase-independent) NEBD-anaphase timer (*Figure 5E*).

## Discussion

### Two Mad2 responses to genome reduplication

Despite a large body of literature describing reduplicated chromosomes in development and disease, the cellular and molecular responses enabling cells to progress through mitosis after genome reduplication have remained essentially unknown. Here, we define two such responses- one that prevents malformation of tissues with polytene chromosomes that persist until anaphase onset, and another (SIRS) that eliminates polyteny before anaphase onset. Both polyteny responses require the conserved mitotic fidelity regulator Mad2, yet Mad2's role in each response is distinct (anaphase delay vs. control of overall mitotic timing). These findings identify new roles for *Drosophila* Mad2, for which few roles have been identified. Further, our findings illuminate a likely recurring role for Mad2 in response to genome reduplication.

In cells with metaphase polytenes (*e.g.* diplochromosomes), our data suggest polytene chromosomes present a challenge for the mitotic spindle, leading to a prolonged period of unattached/tensionless kinetochores. What particular feature of diplochromosomes triggers the SAC wait-anaphase response is unclear. It seems likely that diplochromosome structure is at least partially incompatible with attachment to the spindle. For example, it may be that the outer kinetochores block spindle attachment to the inner kinetochores within a diplochromosome, or it could be that the spindle has trouble generating tension on four kinetochores simultaneously, both of which would trigger the SAC wait-anaphase response. Eventually, the spindle appears able to attach and bi-orient all kinetochores, but the resulting anaphase is frequently error prone- with lagging chromosomes (*Figure 1I*). This result fits with the known ability of cells with erroneous merotelic kinetochore attachments to satisfy the SAC and proceed to anaphase with lagging chromosomes (*Gregan et al., 2011*). Potentially, then, the AuroraB-mediated mechanism that can correct merotely is overwhelmed/inoperable in cells with reduplicated chromosomes (*Cimini et al., 2006*; *Knowlton et al., 2006*). Despite the inability of the SAC to prevent all instances of mitotic errors in cells with polyteny, our data suggest that development of normally diploid tissues with an operable SAC is not noticeably altered by up to 23% ± 4.9% of divisions being error prone tetraploid divisions. Given the conserved nature of SAC signaling, and the widespread occurrence of diplochromosomes in disease, it will be interesting to explore whether the SAC wait-anaphase response is a general mechanism used to enable the expansion of aneuploid cells formed by aberrant genome reduplication.

In contrast to cells with polyteny at anaphase, in cells such as papillar cells, SIRS vastly improves mitotic fidelity. This process does not require the Mad2-dependent wait-anaphase response, but the efficient completion of SIRS before anaphase requires the Mad2-dependent mitotic timer. Little is known about distinct, checkpoint-independent regulation of the Mad2 timer. In the future, papillar cells may prove useful in further study of the timer, given the dependence of SIRS completion on this Mad2 function.

We previously described the error-prone nature of papillar divisions, as well as the high tolerance of this tissue for chromosome mis-segregation. This raises the question of why papillar cells employ SIRS, if papillar aneuploidy is well tolerated (*Schoenfelder et al., 2014*). Based on our study of *HS>fzr* cells, which lack SIRS, we propose that SIRS is required to prevent extreme polytene chromosome mis-segregation events during papillar development, which could result in inviable nullisomic cells. Additionally, we have recently found that papillar cells actively prevent accumulation of micronuclei resulting from broken DNA (*Bretscher and Fox, 2016*). Thus, while mitotic genome-reduplicated cells such as papillar cells do tolerate some degree of aneuploidy, processes such as micronucleus prevention and SIRS may act to ensure a viable degree of mitotic fidelity.

## A model for SIRS

Our results identify SIRS as a spindle-independent chromosome separation process that, remarkably, individualizes polytene chromosomes into recent sister pairs before anaphase. This process is distinct from another spindle-independent chromosome separation process known as C-mitosis, which involves complete sister chromatid separation before anaphase (*Levan, 1938*; *Östergren, 1944*). While future work will determine what differentiates cells capable of SIRS from cells with polytenes that persist until anaphase, our data thus far has examined three layers of polytene chromosome organization that either are or are not eliminated during papillar and *HS>fzr* mitosis, and has pinpointed one of these layers of polytene organization as distinct to cells undergoing SIRS.

The first layer of polytene organization is homolog-homolog pairing. Given that we observe the haploid number of chromosomes after both papillar (*Figure 3B*) and HS>*fzr* (*Figure 1F*) endocycles, it is clear that homologous chromosomes associate within both types of polytene chromosomes by somatic homolog pairing (*Metz, 1916*; *Painter, 1934*). Both mitotic papillar and *HS>fzr* polytenes exhibit un-paired homologs before dividing (*Figure 3B''*, *Figure 1F'*), and this process appears to initiate at centromeres (*Figure 1F*, *Figure 3B*, asterisks, most obvious for the acrocentric X chromosome). Thus, homolog-homolog dissociation is not unique to SIRS. The second layer of polytene organization is interaction between sister chromatid pairs. Importantly, the arrangement of chromatid pairs at metaphase differs between cells that do or do not undergo SIRS. In SIRS-capable (e.g. papillar) cells, only the product of the single most recent round of replication (recent sisters, see nomenclature) remain attached at metaphase whereas the products of previous rounds of replication are no longer attached. In contrast, in SIRS-incapable (e.g. *HS>fzr)* cells, all sister chromatids remain attached. Thus, the separation into chromatid pairs appears to be the critical function of SIRS. Future work can test if this separation requires the Condensin II complex activity during SIRS, which was shown previously to enable partial polytene chromosome dissociation (*Hartl et al., 2008*). The third layer of organization within polytenes are contacts between recent sister chromatid arms. These are equally undone by metaphase in both papillar and HS>*fzr* cells (*Figure 3B''*, *Figure 1F'*) so that, at metaphase, chromatids are only attached at the centromere. This process likely involves the prophase cohesin removal pathway (*Losada et al., 2002*; *Sumara et al., 2002*). Taken together, we conclude the key difference between SIRS-capable (e.g. papillar) and SIRS-incapable (*e.g,* HS>*fzr)* cells is the ability to separate into sister chromatid pairs before metaphase (*Figure 5E*).

We further hypothesize that a key prerequisite to SIRS is the careful regulation of chromosome structure during genome reduplication/endocycles. During the endocycle, papillar cells show no evidence of karyokinesis, which suggests these cells lack a mechanical method of separating chromosomes during the endocycle (*Fox et al., 2010*). However, we speculate that during endocycles, periodic cohesin removal occurs at centromeres after each S-phase. Such cohesin removal would then allow each chromatid to both eliminate its cohesins between its sister from a previous S-phase and then establish cohesins with a new sister during the subsequent S-phase. While alterations in cohesins do not noticeably perturb interphase polytene structure (*Cunningham et al., 2012*; *Pauli et al., 2008*), future work can determine if such endocycle-mediated cohesin regulation confers cells with polytene chromosomes with the ability to undergo SIRS during a later mitosis. Future work can also determine if cohesin regulation differs during endocycles of cells that are destined to later divide. We previously defined features of a distinct pre-mitotic variant of the endocycle, which include centriole retention and the completeness of DNA replication (*Fox et al., 2010*; *Schoenfelder et al., 2014*). Here, we propose that cohesin regulation may be also be distinct during this endocycle variant, and is a key factor to promoting SIRS.

An additional interesting layer of SIRS regulation to explore is how it is triggered, and whether the mitotic timer is an active or passive regulator. Our data suggest NEBD is coincident with SIRS onset, possibly by allowing chromosomes to access some cytoplasmic SIRS regulator, or to initiate SIRS by releasing chromosomes from the nuclear envelope. Regarding the role of the Mad2 timer, it will be interesting to ask if it somehow senses completedness of DNA replication, which may be a pre-requisite for SIRS initiation.

SIRS is likely frequent and conserved. Inspired by classical reports (*Berger, 1938*; *Grell, 1946*; *Holt, 1917*), we found polytene chromosomes are present before polyploid mitosis in *Culex*, but are later apparent as individual chromosomes during mitosis (*Figure 3—figure supplement 1E*). Based on our results, we also suggest that chromosome dispersal in polyploid *Drosophila* ovarian nurse

cells represents an incomplete version of SIRS (*Dej and Spradling, 1999*), especially given that these chromosomes can further separate if mitotic cyclins are experimentally elevated (*Reed and Orr-Weaver, 1997*). In polyploid trophoblasts of some mammalian species, polytene chromosomes separate into numerous bundles of paired chromosomes at the polykaryocyte stage, and thus SIRS may also occur in mammals (*Zybina et al., 1996*; *Zybina et al., 2011*). Similarly, SIRS may eliminate polyteny in some polyploid tumors. One of the first descriptions of polyteny in tumors noted diplochromosomes 'fall apart' <u>before</u> anaphase (*Levan and Hauschka, 1953*). Whole genome duplication is common (~37%) in human tumors (*Zack et al., 2013*). Given the transient nature of polytene chromosomes in mitotic tissues demonstrated here, we suggest future studies of whole genome duplication in cancer models should closely examine the initial mitosis after multiple S-phases to identify potential polytene chromosome origins of tumor aneuploidy. Finally, while our studies agree with the notion that multiple S-phases and polyploidy precede aneuploidy (*Davoli and de Lange, 2012*; *Gordon et al., 2012*), they also underscore the need for aneuploidy-prevention responses including SIRS and the SAC for continued propagation of viable polyploid/aneuploid cells. Future studies can reveal additional SIRS regulation, and other critical genome instability controls in normal or tumorous cells following genome reduplication.

## Materials and methods

### Drosophila genetics

All flies were raised at 25° on standard media (Archon Scientific, Durham, NC). For experiments to measure the length of mitosis larvae or pupae were shifted to 29° for at least 18 hr before dissection. Heat shocks were performed on third instar larvae. Vials were heat shocked in a 37° water bath for 15 min, 30 min, or 60 min. Flybase (flybase.org) describes full genotypes for the following stocks used in this study: *engrailed Gal4* (Bloomington stock 1973); *w1118* (Bloomington stock 3605); *His-2av-GFP* (Bloomington stock 24163); *UAS>GFP.E2f1.1–230, UAS>mRFP1.CycB.1–266* (Bloomington stock 55117). Kyoto DGRC (kyotofly.kit.jp) describes the genotype for the following stock: S/G2/M-Azami (Kyoto stock 109678). The other stocks were generous gifts: *tomato-Cenp-C* (*Althoff et al., 2012*); *HS>fzr* (*Sigrist and Lehner, 1997*); *byn>gal4* (*Singer et al., 1996*); *mad1[1], Df (2R) W45-30n* (*Emre et al., 2011*); *mad2[P]* (*Buffin et al., 2007*), *msl3-GFP* (*Strukov et al., 2011*), *stg>LacZ[4.9]* (Bruce Edgar), *jupiter-GFP* (*Karpova et al., 2006*), and *BubR1-GFP* (*Royou et al., 2010*).

### Mosquito culture

*Culex pipiens* larvae were obtained from Carolina Biological (Burlington, NC). Culturing conditions were as in *Fox et al., 2010*. Larvae were monitored hourly for pupation, and the hindgut was dissected beginning 7 hr post-puparium formation. Antibody staining was as for *Drosophila* tissue.

### Survival analysis

20 wandering 3[rd] instar larvae per replicate of the indicated genotype were placed into a fresh vial with food and then heat shocked for 15 or 30 min. The number of adults that eclosed was counted.

### Chromosome cytology and FISH

Chromosome preparations were based on previous protocols with modifications for the pupal hindgut (*Fox et al., 2010*; *Gatti et al., 1994*). For colcemid treatment, to enrich for metaphase cells, tissue was first incubated in colcemid (Sigma, St. Louis, MO) at 50 µg/ml for 30 min in PBS. For premitotic chromosome spreads with Premature Chromosome Compaction, tissue was incubated in Calyculin A (Cell Signaling Technology, Danvers, MA) at 200 nM in PBS for 30 min, (*Gotoh et al., 1995*; *Miura and Blakely, 2011*). FISH was performed as in *Dej and Spradling, 1999*. BAC clone #BACN04H23 (Chromosome 3L, region 69C3-C8) from the PacMan collection (*Venken et al., 2006*) was labeled using the BioNick labeling system (Invitrogen, Carlsbad, CA). BAC probe signal was amplified through sequential labeling with Peroxidase-labeled Streptavidin followed by the TSA Peroxidase detection kit (Perkin Elmer, Waltham, MA). Imaging was performed on a Zeiss Axio Imager 2 with a 63x oil immersion lens.

## Live imaging

Tissue was dissected and cultured based on previous protocols with modifications for the pupal hindgut (*Fox et al., 2010*; *Prasad et al., 2007*). For colcemid live imaging experiments, pupae were dissected and imaged in media containing 50 µg/ml of colcemid (Sigma, St. Louis, MO) from the initiation of dissection to the first frame was at least 15 min and up to 1 hr. Imaging was performed on a spinning disc confocal (Yokogawa CSU10 scanhead) on an Olympus IX-70 inverted microscope using a 60x/1.3 NA UPlanSApo Silicon oil, 100x/1.4 NA U PlanSApo oil, or a 40x/1.3 NA UPlanFl N Oil objective, a 488 nm and 568 nm Kr-Ar laser lines for excitation and an Andor Ixon3 897 512 EMCCD camera. The system was controlled by MetaMorph 7.7.

## Fixed Imaging

Tissue was dissected in PBS and immediately fixed in 3.7% formaldehyde + 0.3% Triton-X for 15 min. Immunostaining was performed in 0.3% Triton-X with 1% normal goat serum as in *Fox et al., 2010*. The Fluorescent Ubiquitination-based Cell Cycle Indicator (FUCCI) probes (*Zielke et al., 2014*) and mouse anti-Phospho-Histone H3 (ser 10) (Cell Signaling Technology, 1:1000) were used to determine cell cycle stages. Rabbit anti-RFP (MBL, Woburn, MA, 1:500) was used to detect Cenp-C-Tomato Foci. Rabbit anti-DCP1 (Cell Signaling Technology, 1:500) was used to measure cleaved caspases. Mouse anti-γ-tubulin (Sigma, clone GTU-88, 1:1000) was used to detect centrosomes in mitotic cells. TUNEL staining was performed with the in situ cell death detection kit (Roche, Basel, Switzerland) according to the protocol in *Schoenfelder et al., 2014*. Tissue was stained with DAPI at 5 µg/ml. Images were obtained using a Leica SP5 inverted confocal with a 40x or 100x oil objective. Emission was done using a 405 nm diode laser, an argon laser tuned to 488 nm emission, a 561 nm Diode laser, and a 633 HeNe laser.

## Image analysis

All image analysis was performed using ImageJ (*Schneider et al., 2012*). nuclear envelope brightness was calculated by measuring Geminin-Azami intensity or a single cell. The brightness for each cell was normalized from 1 to 0, with 1 being the highest pixel intensity value and 0 being the dimmest value for a single cell across all time-points. Anaphase was determined as the first frame with poleward movement of the kinetochores as evident by Cenp-C-Tomato. Time from NEBD to anaphase was determined as the point from half-maximal Azami signal to anaphase. To calculate kinetochore clustering we closely cropped around the cell of interest for all frames. We then used a thresholding approach to outline each centromere or group of centromeres and stored those as ROIs. We then used those ROIs to measure the average intensity of each centromere pixel and the total area of all the pixels. We reasoned that as centromeres disperse the total area they cover increases and there is a corresponding decrease in fluorescence intensity from each individual point, therefore we divided the average pixel intensity by the area and normalized that on a scale from 1 to 0. To measure symmetry of metaphase centromere alignment (*Figure 5D,D'*), we generated a line plot of each metaphase plate at the frame immediately prior to metaphase. We then generated aggregate plots of each genotype.

## Nomenclature

'N' refers to the haploid number of chromosome sets, while C refers to the haploid DNA content (a diploid cell in G2 is 2N but 4C, a tetraploid cell in G1 is 4N and 4C). For chromosomes we use 'homolog' to distinguish maternally and paternally contributed chromosomes of the same chromosome type. All chromatids of the same homolog are considered 'sisters.' We use 'recent sister' to refer to two chromatids that are the product of the most recent S-phase. 'Polytene' refers to the chromosome state of any cell formed by genome reduplication that has not fully separated its chromosomes into recent sisters. In a polyploid/polytene cell with somatic homolog pairing, polytene cells exhibit the haploid number of distinguishable chromosomes, whereas in a cell without homolog pairing, this number doubles. 4 chromatids conjoined at centromeres are 'diplochromosomes'. Please note that, using metaphase spreads, the presence/absence of diplochromosomes as well as the number of individual chromatids can only be scored when chromosomes are significantly condensed as in mitosis or in the presence of Calyculin A. We use the term "endocycle" to refer to any cell cycle involving successive genome reduplication without any sign of M-phase. We note the use

in the literature of terms such as 'endoreduplication', 're-replication', and 'endoreplication' to often refer to the same phenomenon.

## Acknowledgements

The following kindly provided reagents used in this study: The Bloomington Stock Center, the Vienna Drosophila Resource Center, the Kyoto Drosophila Genetic Resource Center, Roger Karess (U Paris), Christian Lehner (U Zurich), Mitzi Kuroda (Harvard University), and Bill Sullivan (UC Santa Cruz). We thank David MacAlpine, Eda Yildirim, Beth Sullivan, Joshua Bembenek, and Fox lab members for valuable comments on the manuscript. This project was supported by both NIGMS grant GM118447 and a Pew Scholar Award (Pew Charitable Trusts) to DF.

## Additional information

### Funding

| Funder | Grant reference number | Author |
| --- | --- | --- |
| Pew Charitable Trusts | | Donald T Fox |
| National Institutes of Health | GM118447 | Donald T Fox |

The funders had no role in study design, data collection and interpretation, or the decision to submit the work for publication.

### Author contributions
BMS, DTF, Contributed to conception and design, Data acquisition, Analysis of data, Drafting/revising the article

### Author ORCIDs
Donald T Fox, http://orcid.org/0000-0002-0436-179X

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
