## [Decision Letter]

Thank you for submitting your article "Distinct Responses To Reduplicated Chromosomes Require Distinct Mad2 Responses" for consideration by *eLife*. Your article has been favorably evaluated by K VijayRaghavan (Senior editor) and three reviewers, one of whom, Yukiko Yamashita, is a member of our Board of Reviewing Editors, and another is Giovanni Bosco.

The reviewers have discussed the reviews with one another and the Reviewing Editor has drafted this decision to help you prepare a revised submission.

Summary of the work:

In this study Stormo and Fox use two in vivo systems of *Drosophila* to ask how endoreduplicated cells cope with eventually segregating polytene chromosomes. First they drive endoreduplication in *Drosophila* tissues that are normally diploid by a heat shock inducible *fzr* gene. Second, they examine rectal papilla cells that normally undergo mitosis after having reduplicated their genomes. This study represents a significant new contribution to the way we think about how chromosome organizational states (e.g. polytenes) are normally modified, while it also reveals a very important new checkpoint-independent function of the Mad2 gene. The study is not particularly mechanistic and remains essentially descriptive, but it nevertheless makes an interesting contribution to the field.

An important and novel finding of this study is that the authors elegantly show distinct roles for the conserved Mad2 mitotic checkpoint signaling gene. In addition to its well-known spindle assemble checkpoint (SAC) function, the authors also show that cells containing polyploid cells that have persistent polytene chromosome structures also benefit from a Mad2 dependent mitotic timer function. This Mad2 checkpoint-independent mitotic timer function, it is proposed, promotes separation into recent sister chromatids (SIRS), and presumably disrupts other interactions, e.g. homolog pairing that tend to be strong and ubiquitous in diploid and endoreduplicating *Drosophila* cells.

Overall, all reviewers agreed that this work represents a significant progress in the field that warrants publication in *eLife*. The following comments must be addressed, although the reviewers do not necessarily believe the authors need additional experiments.

Suggested revisions:

There is no experimental exploration of what is mechanistically happening to the polytene chromosomes that drives them to separate and individualize into recent sister pairs. Presumably cohesion is keeping the most recent sisters cohesed (or maybe catenations or under-replication, etc.?) in anticipation of metaphase. However, what is eliminating arm/centromeric cohesion and disrupting homolog interactions before the metaphase-to-anaphase transition? Obvious candidates would be processes targeting cohesins, but Top-2 or condensin II driven resolution of catenations and/or homolog pairing are other possibilities that could easily have been tested (Top-2 in particular since pharmacological inhibition could have been achieved in combination with live imaging, etc.). Can the authors speak to what is the most likely scenario and what supporting data there may be?

The authors state that chromosome missegregation in cells undergoing the first division after induced reduplication is mainly due to the persistence of "diplochromosomes", i.e. homologous chromosomes completely separated but still conjoined at their centromeres. This is rather clearly apparent in the metaphase spread shown in Figure 1'. However, the presence of "diplochromosomes" is not as clearly documented in the live imaging reported in Figure 1 and Figure 1—figure supplement 2. In addition, how can authors exclude that the observed segregation defects are mainly due to the presence of multipolar spindles? Actually, the presence of frequent tripolar divisions has been reported by Schoenfelder et al. (2014) in 8N papillar cells.

Based on the evidence that loss of Mad2 abolishes the lengthened period of metaphase caused by diplochromosomes, authors conclude that a wait-anaphase signal is activated in the presence of diplochromosomes. However, no movie is shown to support this conclusion. In addition, they only indirectly demonstrate that the absence of a delay in the metaphase-anaphase progression corresponds to an increase of defective mitoses, through the analysis of the survival rate and of the developmental defects observed in *fzr* overexpressing *mad2* null animals. Can they quantitate the frequency of aberrant anaphases in the induced polytenic cells of *mad2* null animals? Also, related to this point, if Mad1 is dispensable and they claim Mad2's "checkpoint-independent role", they should not call "Mad2 SAC" (it is simply a Mad2-depentent novel mechanism, not "SAC" per se).

The authors first state that papillar cells do not have quadruplochromosomes and then use a substantial part of the paragraph entitled "Polyteny response 2: Separation…" to demonstrate that polytene chromosomes can be visualized in these cells. This is a bit confusing. The text could be shortened to help the reader to get the point, that is the fact that in papillar cells polytene chromosomes are formed and have the ability to progressively separate into sisters prior to metaphase. Here again, the cytology on fixed material looks more beautiful and convincing than the live imaging analysis. However, the authors should explain why in the metaphase shown in Figure 3" four of the green (presumably third) chromosomes show a clear pericentric inversion. Are they balancers? In the same metaphase one of the green chromosomes exhibits a chromatid break. Isn't it supposed to be a wild type metaphase?

It is not clear how *mad2* null animals fail to complete SIRS before anaphase, as the video frames in Figure 5 refer only to a wild type cell. Representative frames from a video of a *mad2* null papillar cell mitosis should be reported in the figure.

Can the authors please clarify in the text and their model how they know that metaphase chromatids in SIRS cells are the most recent sister pairs? The speculation (subsection “A model for SIRS”, second paragraph) of cohesin removal between old sisters so that new cohesins could establish cohesion between the most recent sisters is a reasonable idea, however this would be greatly strengthened by somehow showing that SIRS cells entering metaphase indeed have pairs of new/recent sisters as opposed to a random arrangement of sister pairs after the second S-phase. Are there data in the literature that cohesins can be removed in endoreduplicating cells?

It would be of interest and helpful to a broader readership to further elaborate in the beginning of the manuscript what physical connections between polytene sisters/homologs might exist and which of these are analogous in mammalian cells that may display "polytene-like" structures. Although it is specified in the first paragraph of the Introduction that "polytene" refers to the persistence of chromatid associations, most readers will assume that especially in *Drosophila* "polytene" also refers to homolog-homolog pairing. Can the authors clarify this? This may be difficult to do since we don't know the actual nature of the physical interactions that form/maintain *Drosophila* polytenes (likely not cohesins) and how similar they are to analogous chromosome structures in non-*Drosophila* cells. Nevertheless, this might be important to emphasize since it may be challenging for a non-*Drosophila*, non-polytene-centric viewpoint to understand the relevance of this new and exciting Mad2 driven SIRS mechanism. Similarly, in the third paragraph of the Introduction, the description about "endocycled cells" and "polyteny" is somewhat unclear. First of all, it is unclear how one can distinguish polytene chromosomes from regular chromosomes (containing only single sister chromatid). For example, how one can surely tell 4N vs. 8N? Are diplochromosomes from 4N (N + N + N + N) easily distinguishable from 2N+2N+2N+2N from 8N chromosomes, which might look diplochromosomes but each containing double the amount of DNA?

Is it possible that an important difference between SIRS compatible cells and SIRS non-compatible cells is some degree of DNA underreplication in polytene cells and their ability to resolve/finish this DNA replication? Is the Mad2 timer sensing unreplicated DNA – perhaps at/around the centromere? It would be interesting and informative for the authors to consider this possibility in their Discussion as it might help readers frame some of the important questions raised by this study in the context of issues relevant to what may actually be happening during the process of achieving SIRS.

In the Discussion, two possibilities are considered for why polyteny is incompatible with spindle attachment to kinetochores. However, is it possible that paired arms of sisters and/or persistent homolog pairing is also incompatible with mitotic progression? Are there data in the literature or from the authors' own work that speaks to this either way?

The use of the phrase "eliminates polytene chromosomes" (such as in the Discussion subsection “A model for SIRS”, first paragraph and other variants thereof) implies that chromosomes themselves are destroyed or lost. I think the authors mean that the polytene organization of the chromosomes is resolved such that sisters and homologs are individualized. If this is a correct interpretation of what the authors mean, then they may wish to use different words or phrase to clearly indicate that chromosomes are not being "eliminated" per se, but instead it's the polytene organization of chromosomes that is being eliminated.

The authors state "consistent with previous work" (or alike) in many places. This gives false impression that this work lacks novelty: if one carefully reads the paper and goes to the references, the references are often talking about the phenomenon in a different context (e.g. previous work showed something in diploid cells, and the authors showed a similar phenomenon in tetraploid cells). The authors should make it clear in what aspect it is consistent with the previous work, and what is novel in this work.

---

## [Author Response]

Suggested revisions:

There is no experimental exploration of what is mechanistically happening to the polytene chromosomes that drives them to separate and individualize into recent sister pairs. Presumably cohesion is keeping the most recent sisters cohesed (or maybe catenations or under-replication, etc.?) in anticipation of metaphase. However, what is eliminating arm/centromeric cohesion and disrupting homolog interactions before the metaphase-to-anaphase transition? Obvious candidates would be processes targeting cohesins, but Top-2 or condensin II driven resolution of catenations and/or homolog pairing are other possibilities that could easily have been tested (Top-2 in particular since pharmacological inhibition could have been achieved in combination with live imaging, etc.). Can the authors speak to what is the most likely scenario and what supporting data there may be?

As requested, we performed the reviewers’ requested experiment of live imaging papillar cells during SIRS in the presence of etoposide, a Top-2 pharmacological inhibitor. This experiment provided the insight that Top-2 is not required for SIRS at centromeres. However, Top-2 inhibition does result in anaphase DNA bridges in all cell types tested, including papillar cells and diploid wing disc cells. In papillar cells these bridges could be the result of either a) failure of SIRS along chromosome arms, b) failure of the expected role of Top-2 in resolving sister chromatid catenation, or c) both a and b. As Top-2 inhibition prevents the examination of chromosome structure by metaphase spreads to differentiate possibilities a, b, and c, future work beyond a reasonable timeframe for this manuscript is required to tease apart whether or not Top-2 plays a non-centromeric role in SIRS. We provide data on this experiment in Figure 610.7554/eLife.15204.030Author Response Image 1.**DOI:**
http://dx.doi.org/10.7554/eLife.15204.030

Regarding further investigation of the distinct regulation of: arm cohesins, centromeric cohesins, catenation/topoisomerases, under-replication, homolog pairing, and condensins during: endoreduplication, SIRS, and diplochromosome separation, in-depth exploration of these mechanisms in each context are clearly interesting questions that we will explore but are beyond the scope of this study.

The authors state that chromosome missegregation in cells undergoing the first division after induced reduplication is mainly due to the persistence of "diplochromosomes", i.e. homologous chromosomes completely separated but still conjoined at their centromeres. This is rather clearly apparent in the metaphase spread shown in Figure 1'. However, the presence of "diplochromosomes" is not as clearly documented in the live imaging reported in Figure 1 and Figure 1—figure supplement 2.

Thank you to the reviewers for allowing us to clarify the appearance of diplochromosomes in live imaging experiments. Diplochromosomes are certainly the most obvious by metaphase spread analysis, which is the technique that has been used for decades to examine these structures. However, we found that we can also observe diplochromosomes during anaphase in live imaging. We expand here on their distinct appearance in live imaging. In live imaging of anaphases of first division *HS>fzr* cells, we observe chromatid quartets, which consist of four clear centromeres and associated chromosome arms in very close proximity. These quartets are often seen lagging behind other chromosomes at anaphase, as depicted in Figure 1. The appearance of quartets is exactly what one would expect if a diplochromosome were to be delayed in its segregation, as our results in Figure 2 suggest to be the case.

We never observe quartets in live imaging of diploid anaphases or papillar anaphases. Further, please note that these quartet structures are *not* seen in the second division (Figure 1—figure supplement 1). From the reviewers’ comment, it’s clear that we did not sufficiently highlight the difference between Figure 1 (diplochromosomes present) and Figure 1 (diplochromosomes absent). We have thus revised the text to make this point clearer. In summary, due to both protocol differences (fixed spreads vs. live imaging) and temporal differences (metaphase vs. anaphase), diplochromosomes are not morphologically identical to their appearance in metaphase spreads, but are clearly recognizable in live images of the first *HS>fzr* division. We have revised the text to make this point clearer.

In addition, how can authors exclude that the observed segregation defects are mainly due to the presence of multipolar spindles? Actually, the presence of frequent tripolar divisions has been reported by Schoenfelder et al. (2014) in 8N papillar cells.

As requested, we now provide additional data regarding multipolar divisions. As the reviewers point out, we previously documented tripolar divisions after endocycles in papillar cells. However, we would like to point out that these divisions are in fact infrequent (~7% of all endocycled papillar cells- Schoenfelder et al., 2014), due to the inefficient duplication of centrosomes during papillar endocycles. We have since examined whether centrosomes duplicate, and to what extent, in *HS>fzr* cells after heat shock. Much like papillar cells, we find very few (7%) *HS>fzr* cells contain extra centrosomes at the first division, as assayed by Gamma Tubulin antibody staining. Accordingly, 0 out of 65 first division *HS>fzr* cells undergo multipolar division. These data are now included in Figure 1—figure supplement 1 (panels F and G) and we have added additional text that addresses this point raised by the reviewers. Thus, we conclude that diplochromosomes – not extra centrosomes/multipolar spindles – are the main cause of chromosome segregation defects in *HS>fzr* animals.

Based on the evidence that loss of Mad2 abolishes the lengthened period of metaphase caused by diplochromosomes, authors conclude that a wait-anaphase signal is activated in the presence of diplochromosomes. However, no movie is shown to support this conclusion. In addition, they only indirectly demonstrate that the absence of a delay in the metaphase-anaphase progression corresponds to an increase of defective mitoses, through the analysis of the survival rate and of the developmental defects observed in fzr overexpressing mad2 null animals. Can they quantitate the frequency of aberrant anaphases in the induced polytenic cells of mad2 null animals? Also, related to this point, if Mad1 is dispensable and they claim Mad2's "checkpoint-independent role", they should not call "Mad2 SAC" (it is simply a Mad2-depentent novel mechanism, not "SAC" per se).

As requested, we now include multiple new movies/figure panels regarding the wait-anaphase response to diplochromosomes. 1) We now include panel C of Figure 2—figure supplement 1, and movies for Figure 2—figure supplements 3 and 4, which show that BubR1, a component of the wait-anaphase signal, persists much longer on some chromosomes in cells with diplochromosomes. 2) We now include data on the frequency of aberrant anaphases in *HS>fzr* mad 2 polytene anaphases (Figure 2—figure supplement 1). Further, we removed all references to a “Mad2 SAC” from the text.

The authors first state that papillar cells do not have quadruplochromosomes and then use a substantial part of the paragraph entitled "Polyteny response 2: Separation…" to demonstrate that polytene chromosomes can be visualized in these cells. This is a bit confusing. The text could be shortened to help the reader to get the point, that is the fact that in papillar cells polytene chromosomes are formed and have the ability to progressively separate into sisters prior to metaphase. Here again, the cytology on fixed material looks more beautiful and convincing than the live imaging analysis.

Thank you to the reviewers for this suggestion on how to improve the text. We have made these requested revisions. Regarding the differences between fixed and live material, as discussed in point #2 of our response there are inherent differences in chromosome appearance between live and fixed preparations. We certainly agree that fixed chromosome spreads are the most beautiful way to examine chromosome structure. Live imaging of course has its own benefits, such as enabling us to see the fate of polytene chromosomes over time.

However, the authors should explain why in the metaphase shown in Figure 3" four of the green (presumably third) chromosomes show a clear pericentric inversion. Are they balancers? In the same metaphase one of the green chromosomes exhibits a chromatid break. Isn't it supposed to be a wild type metaphase?

Thank you to the reviewers for these questions. You are absolutely correct on both fronts. We find that inversions on balancer chromosomes locally disrupt the tight homolog-homolog pairing in papillar polytenes, which is of course expected given the long history of study of chromosome pairing in polyploid cells. We now discuss this in the text and add Figure 3—figure supplement 1 panel B that highlights the effect of balancers on papillar chromosome structure. With regards to DNA breakage, as we discuss below and as we previously published (Fox et al. 2010) this *is* a naturally occurring feature of wild type papillar chromosomes. How papillar cells respond to naturally occurring DNA breaks is the subject of a separate manuscript currently in press at Developmental Cell. We now note this in the legend of Figure 3.

It is not clear how mad2 null animals fail to complete SIRS before anaphase, as the video frames in Figure 5 refer only to a wild type cell. Representative frames from a video of a mad2 null papillar cell mitosis should be reported in the figure.

As requested, we have altered figure Figure 5, which more clearly shows the failure of SIRS in *mad2* null papillar mitosis.

Can the authors please clarify in the text and their model how they know that metaphase chromatids in SIRS cells are the most recent sister pairs? The speculation (subsection “A model for SIRS”, second paragraph) of cohesin removal between old sisters so that new cohesins could establish cohesion between the most recent sisters is a reasonable idea, however this would be greatly strengthened by somehow showing that SIRS cells entering metaphase indeed have pairs of new/recent sisters as opposed to a random arrangement of sister pairs after the second S-phase.

As requested, we provide additional information to the reviewers regarding the question of papillar chromatid arrangement. We do in fact have compelling evidence that papillar chromatids are arranged in a conventional recent-sister pair configuration. This evidence comes from another study in our lab (currently in press at Developmental Cell) that explored the frequent appearance of broken chromosomes in papillar cells. These breaks are frequently at the same place on adjacent chromatids in a sister pair, which could best be explained by a break that occurred prior to or during S-phase, which was then propagated to both halves of the most recent sister pair. These data lead us to conclude that recent sisters remain in a conventional arrangement in papillar cells following endocycles. We provide evidence of our chromosome breakage data in Figure 6

Are there data in the literature that cohesins can be removed in endoreduplicating cells?

Thank you to the reviewers for this question. We are not aware of any study that specifically examined whether cohesins are dynamic during the period of endoreduplication. However, we do cite studies from the Kassis and Nasmyth laboratories showing that cohesins are dynamic in polytene chromosomes from larval salivary glands (which may or may not have finished endoreduplication). In the near future, we plan to undertake our own studies and to examine whether cohesins, if dynamic in papillar cells, are similarly dynamic in other polytene cells that do not undergo future divisions. It is possible that cohesin regulation is distinct in cells that are never destined to undergo SIRS.

It would be of interest and helpful to a broader readership to further elaborate in the beginning of the manuscript what physical connections between polytene sisters/homologs might exist and which of these are analogous in mammalian cells that may display "polytene-like" structures. Although it is specified in the first paragraph of the Introduction that "polytene" refers to the persistence of chromatid associations, most readers will assume that especially in Drosophila "polytene" also refers to homolog-homolog pairing. Can the authors clarify this? This may be difficult to do since we don't know the actual nature of the physical interactions that form/maintain Drosophila polytenes (likely not cohesins) and how similar they are to analogous chromosome structures in non-Drosophila cells. Nevertheless, this might be important to emphasize since it may be challenging for a non-Drosophila, non-polytene-centric viewpoint to understand the relevance of this new and exciting Mad2 driven SIRS mechanism. Similarly, in the third paragraph of the Introduction, the description about "endocycled cells" and "polyteny" is somewhat unclear. First of all, it is unclear how one can distinguish polytene chromosomes from regular chromosomes (containing only single sister chromatid). For example, how one can surely tell 4N vs. 8N? Are diplochromosomes from 4N (N + N + N + N) easily distinguishable from 2N+2N+2N+2N from 8N chromosomes, which might look diplochromosomes but each containing double the amount of DNA?

As requested, we expanded the Introduction per the reviewers’ suggestion. The reviewers also raise a number of important points about the often challenging and non-uniform use of nomenclature related to polyploidization cell cycles, polyploidy, and polytene chromosomes – we’ve expanded the nomenclature section of the Methods to more clearly and extensively define polytene structure and when it is possible to distinguish diplochromosomes and numbers of chromatids.

Is it possible that an important difference between SIRS compatible cells and SIRS non-compatible cells is some degree of DNA underreplication in polytene cells and their ability to resolve/finish this DNA replication? Is the Mad2 timer sensing unreplicated DNA – perhaps at/around the centromere? It would be interesting and informative for the authors to consider this possibility in their Discussion as it might help readers frame some of the important questions raised by this study in the context of issues relevant to what may actually be happening during the process of achieving SIRS.

Thank you to the reviewers for these interesting suggestions. We now discuss them in the text.

*In the Discussion, two possibilities are considered for why polyteny is incompatible with spindle attachment to kinetochores. However, is it possible that paired arms of sisters and/or persistent homolog pairing is also incompatible with mitotic progression? Are there data in the literature or from the authors' own work that speaks to this either way?*

Thank you for giving us the opportunity to expand on our discussion of homolog pairing. While we understand why one would wonder about pairing in this context, we find the reviewers’ model to be unlikely, based on both our own data and on previously published data. First, we find that homologous chromosomes are not paired when *HS>fzr* cells are in mitosis (see Figure 1’). Second, our finding is consistent with previous work by others in diploid cells, such as from the Sedat lab (Fung et al. 1998) showing that homolog pairing is disrupted by mitosis. We have made this point clearer in the text. Thus, we consider it unlikely that homolog pairing influences polytene chromosome segregation in *HS>fzr* cells.

The use of the phrase "eliminates polytene chromosomes" (such as in the Discussion subsection “A model for SIRS”, first paragraph and other variants thereof) implies that chromosomes themselves are destroyed or lost. I think the authors mean that the polytene organization of the chromosomes is resolved such that sisters and homologs are individualized. If this is a correct interpretation of what the authors mean, then they may wish to use different words or phrase to clearly indicate that chromosomes are not being "eliminated" per se, but instead it's the polytene organization of chromosomes that is being eliminated.

Thank you – we have revised the text to clarify this point.

The authors state "consistent with previous work" (or alike) in many places. This gives false impression that this work lacks novelty: if one carefully reads the paper and goes to the references, the references are often talking about the phenomenon in a different context (e.g. previous work showed something in diploid cells, and the authors showed a similar phenomenon in tetraploid cells). The authors should make it clear in what aspect it is consistent with the previous work, and what is novel in this work.

We have re-reviewed the literature and revised/qualified the text where necessary. Thank you for noting the novelty of our work.